# A global meta-analysis of yield stability in organic and conservation agriculture

Samuel Knapp [1,2] & Marcel G.A. van der Heijden [1,3]

One of the primary challenges of our time is to enhance global food production and security. Most assessments in agricultural systems focus on plant yield. Yet, these analyses neglect temporal yield stability, or the variability and reliability of production across years. Here we perform a meta-analysis to assess temporal yield stability of three major cropping systems: organic agriculture and conservation agriculture (no-tillage) vs. conventional agriculture, comparing 193 studies based on 2896 comparisons. Organic agriculture has, per unit yield, a significantly lower temporal stability (−15%) compared to conventional agriculture. Thus, although organic farming promotes biodiversity and is generally more environmentally friendly, future efforts should focus on reducing its yield variability. Our analysis further indicates that the use of green manure and enhanced fertilisation can reduce the yield stability gap between organic and conventional agriculture. The temporal stability (−3%) of no-tillage does not differ significantly from those of conventional tillage indicating that a transition to no-tillage does not affect yield stability.

[1] Plant-Soil-Interactions, Research Division Agroecology and Environment, Agroscope CH 8046 Zurich, Switzerland. [2] Chair of Plant Nutrition, Department of Plant Sciences, Technical University of Munich, Emil-Ramann-Strasse 2, 85354 Freising, Germany. [3] Department of Plant and Microbial Biology, University of Zurich, CH 8008 Zurich, Switzerland. These authors contributed equally: Samuel Knapp, Marcel G.A. van der Heijden. Correspondence and requests for materials should be addressed to M.G.A.v.d.H. (email: marcel.vanderheijden@agroscope.admin.ch)

Continuing population and consumption growth will mean that the global demand for food will increase for at least another 40 years[1–3]. It is, thus, a key challenge to enhance food security. This requires a multifaceted global strategy at all scales, from farm to global level, including factors, such as reducing food production limits, reducing temporal yield variability, reducing food waste and changing diets[4]. Moreover, stable food production will be a greater challenge under a changing and less predictable climate[5].

In addition to the challenges of enhancing food security, there is a growing recognition that agriculture must produce more sustainably. Intensive conventional agriculture has more than tripled yield in the last century[1–3]. However the use of pesticides and mineral fertilisers in conventional agriculture often has a negative impact on the environment through decreasing biodiversity, pollution and eutrophication of water, and degrading soil quality[2–4]. Thus, there is the challenge to simultaneously enhance global food security and to reduce the environmental impact of agriculture.

Organic farming and conservation agriculture are two alternatives to conventional agriculture and are often promoted as more environmentally friendly practices[6–8]. Organic agriculture is defined as having no synthetic inputs (no synthetic pesticides and no mineral fertilisers)[9,10], and a range of studies show that organic farming enhances biodiversity and has reduced environmental impact[6,8,11]. Conservation agriculture represents a set of three crop management principles: (A) direct planting of crops with minimum soil disturbance (that is, reduced or no-tillage), (B) permanent soil cover by crop residues or cover crops, and (C) crop rotation[12]. Several studies indicate that conservation agriculture has a positive effect on soil quality and a range of soil biota[7,13,14].

So far, studies comparing organic or conservation agriculture with conventional agriculture have tested whether organic agriculture or conservation agriculture differ in yield, biodiversity or environmental services compared to conventional agriculture. However, an important issue that is relevant for the discussion on food security is that of yield stability (i.e., the variability of yield across years). So far, it has not been tested whether yield stability in organic and conservation agriculture differs from that in conventional agriculture.

The concept of yield stability was originally developed in plant breeding[15,16], but in recent years it has also received increased interest from ecologists, especially in relation to the stability of ecosystem functioning[17,18]. Yield stability can be measured in various ways[16]. One way to measure temporal yield variability is the standard deviation of yield across years. We refer to this as the absolute stability. However, this measure does not account for the differences in yield. Hence, various investigators have calculated the coefficient of variation, which divides the variability across years (expressed as standard deviation) by the mean yield over the same period[17–20]. In order to distinguish from absolute stability, we refer to this as relative stability. Different from absolute stability, relative stability is scaled per unit yield produced. This means that both the variability across years and the mean yield level influence relative yield stability (e.g., a treatment with reduced yield but equal absolute stability (standard deviation) has a reduced relative yield stability (greater coefficient of variation) because the amount of variation per unit yield is higher. Many factors can cause yield of crop species to vary across years, including differences in precipitation, temperature, pest outbreaks, weed pressure, soil fertility, soil structure and agricultural management[10].

Two recent meta-analyses compared the yield of conventional agriculture with organic agriculture and conservation agriculture. A study by Ponisio et al.[21], building upon Seufert et al.[22] and de Ponti et al.[23], compared 1071 paired yield observations of 115 studies and showed that organically managed fields have on average 19.2% less yield compared to conventionally managed fields. It was further observed that the yield gap between organic and conventional agriculture depends on crop species, and it was lower when both systems used crop rotations or received similar amounts of fertiliser. Another recent meta-analysis by Pittelkow et al.[12] compared no-tillage, the original and central concept of conservation agriculture, with conventional tillage and observed that no-tillage on average reduced yields by 5.7% compared to conventional tillage. The effects were variable, depended on crop species[12] and under certain conditions no-tillage produced equivalent or even greater yields than conventional tillage.

We applied a meta-analysis procedure using the datasets by Ponisio et al.[21] and Pittelkow et al.[12] and compared temporal yield stability of (A) organic vs. conventional agriculture and, (B) no-tillage vs. conventional tillage. We used 191 studies (39 studies from Ponisio et al.[12] and 154 studies from Pittelkow et al.[21]) resulting in 532 multiple year observations that were based on 2896 comparisons. We demonstrate that relative yield stability of organic agriculture, assessed per unit yield produced, is significantly lower compared to conventional agriculture. Moreover absolute stability (i.e. the temporal variation in plant yield without correcting for yield level) did not differ between organic and conventional agriculture. Our analysis further indicates that enhanced fertilisation and the application of green manure can help to reduce the yield stability gap with conventional agriculture and reduce relative yield stability in organic agriculture. We further show that no-tillage and conventionally tilled systems have similar yield stability, especially in dry climates and on fields with residue retention and crop rotation.

## Results

**Yield stability of organic and conventional agriculture**. We used the dataset of Ponisio et al.[12] to compare temporal yield stability of organic and conventional agriculture. Our analysis demonstrates that the relative yield stability (i.e., yield stability per unit yield produced) in conventionally managed fields was, averaged across all crops, 15% [2–30%] higher compared to organically managed fields, and this difference was significant (Fig. 1). A closer look at the data further confirmed this, and out of the 165 multiple year comparisons (observations) in the dataset, 79% (131 observations) had higher relative stability in conventionally managed fields (Fig. 2). We observed no significant difference in absolute stability between organic and conventional agriculture (Fig. 1) demonstrating that the overall temporal variability in yield, independent of yield level, was similar between organic and conventional agriculture.

We observed a significantly increased relative yield stability under conventional management for two (soybean and barley) out of five crop species for which enough data (>10 comparisons) were available (Fig. 1). Interestingly, a significant difference in absolute stability (i.e., not corrected for yield level) was only observed for soybean. The absolute stability of soybean was higher in conventionally managed fields compared to organically managed fields. Results for many other crop species were highly variable (Supplementary Table 1) and should be interpreted carefully because few data (often only one or two comparisons) were available.

We evaluated the effects of other factors on yield stability, and our analysis indicated that the increased relative and absolute yield stability of conventional management was related to differences in N-fertilisation (Fig. 3). If organically and conventionally managed fields received similar amounts of nitrogen fertiliser, relative yield stability did not vary significantly

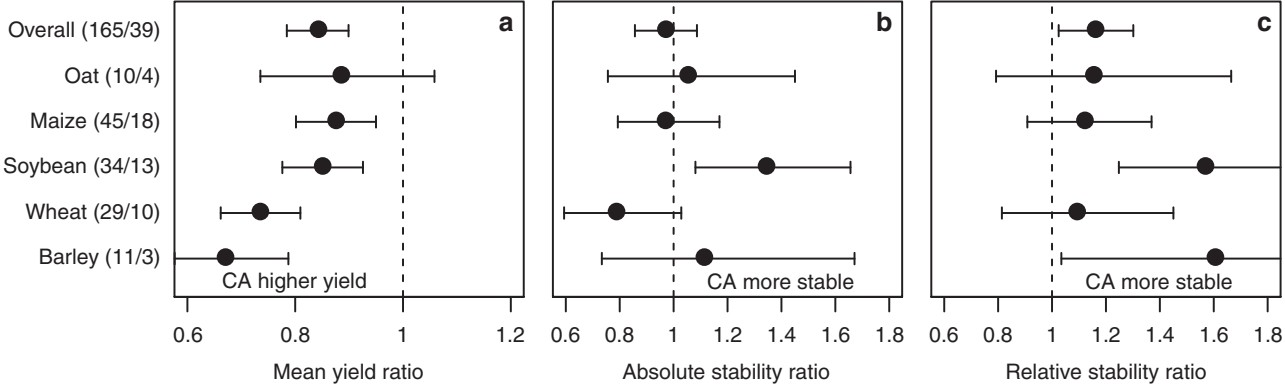

**Fig. 1** Yield and yield stability comparing organic and conventional agriculture. **a** Mean yield ratio. **b** Absolute stability ratio. **c** Relative stability ratio for organic (OA) vs. conventional (CA) agriculture for all crops (Overall) and for crops, for which at least 10 observations were available. Numbers in parentheses denote the number of observations and studies. A ratio of 1 means that there is no difference between organic and conventional managed systems while values <1 indicate higher yield for conventional agriculture. For both stability measures a ratio >1 indicate greater absolute and relative stability for conventional agriculture. Values are mean effect sizes with 95% confidential intervals. Mean yield or stability were deemed significantly different between organic and conventional agriculture if the 95% confidential intervals of the ratios did not overlap one

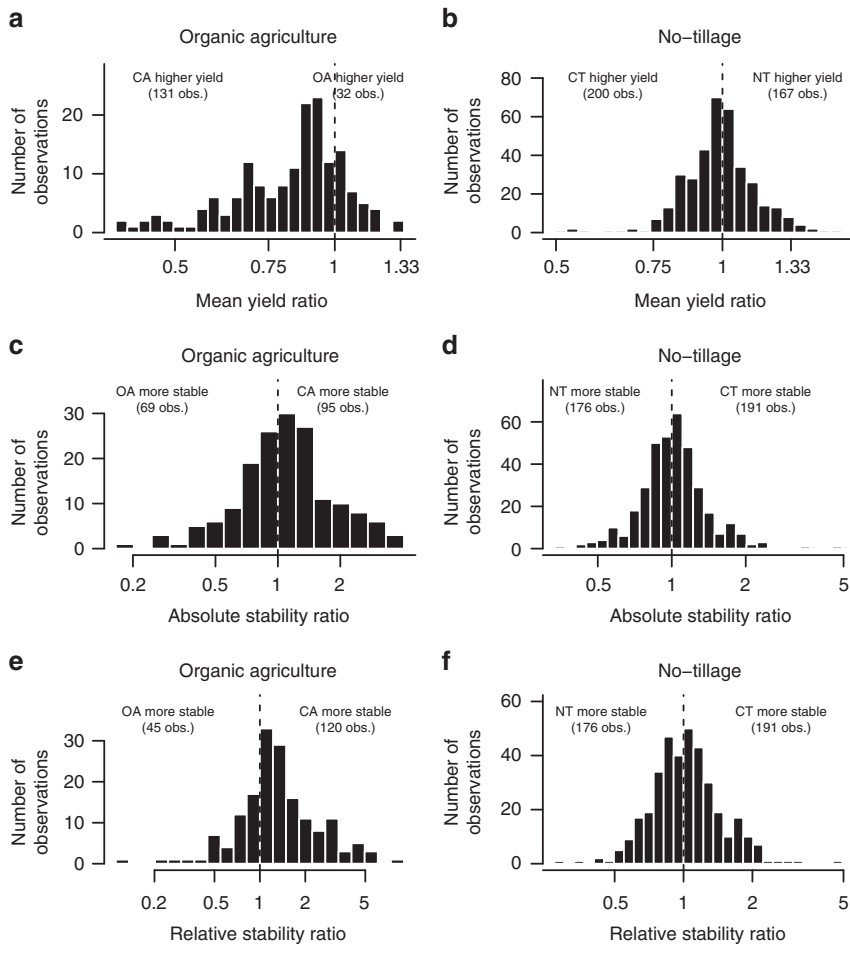

**Fig. 2** Histograms for yield and yield stability ratios. **a**, **b** Mean yield ratios. **c**, **d** Absolute stability ratios. **e**, **f** Relative stability ratios for the dataset comparing organic (OA) and conventional agriculture (CA) (**a**, **c**, **e**) and the dataset comparing no-tillage (NT) and conventional tillage (CT) (**b**, **d**, **f**), respectively. The ratios on the x-axis are on the ln-scale

between both management systems; although it was still lower (9%) in organically managed systems. However, if organically managed fields received less nitrogen, the relative yield stability was much lower (42% [−11% to −81%]) compared to conventionally managed fields. This indicates that the increased relative stability of conventionally agriculture is, in part, due to higher fertilisation levels and related to the higher yield. Still, even with equal amounts of nitrogen fertilisation, organic agriculture had a significantly lower yield (12% [−2% to −21%]); although this difference was less than for the overall dataset where it was

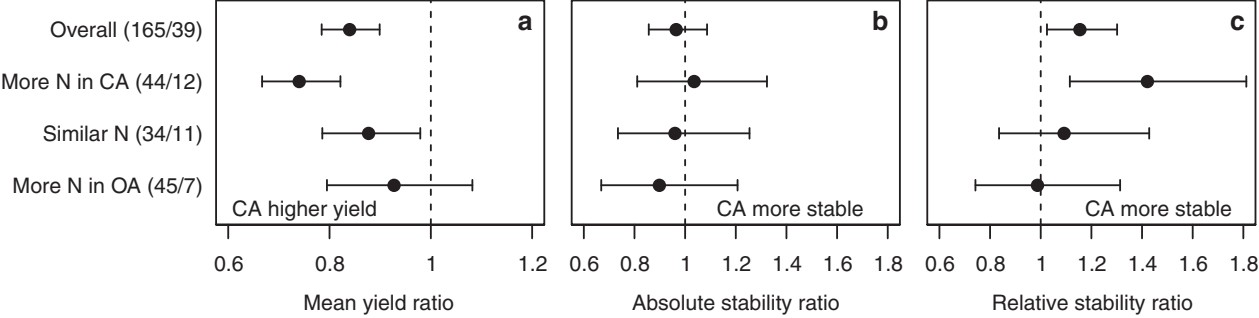

**Fig. 3** Effect of nitrogen input on yield and yield stability comparing organic and conventional agriculture. **a** Mean yield ratio. **b** Absolute stability ratio. **c** Relative stability ratio for organic (OA) vs. conventional (CA) agriculture for different levels of nitrogen input. Numbers in parentheses denote the number of observations and studies. A ratio of 1 means that there is no difference between organic and conventional managed systems while values <1 indicate higher yield for conventional agriculture. For both stability measures a ratio >1 indicate greater absolute and relative stability for conventional agriculture. Values are mean effect sizes with 95% confidential intervals. Mean yield or stability were deemed significantly different between organic and conventional agriculture if the 95% confidential intervals of the ratios did not overlap one

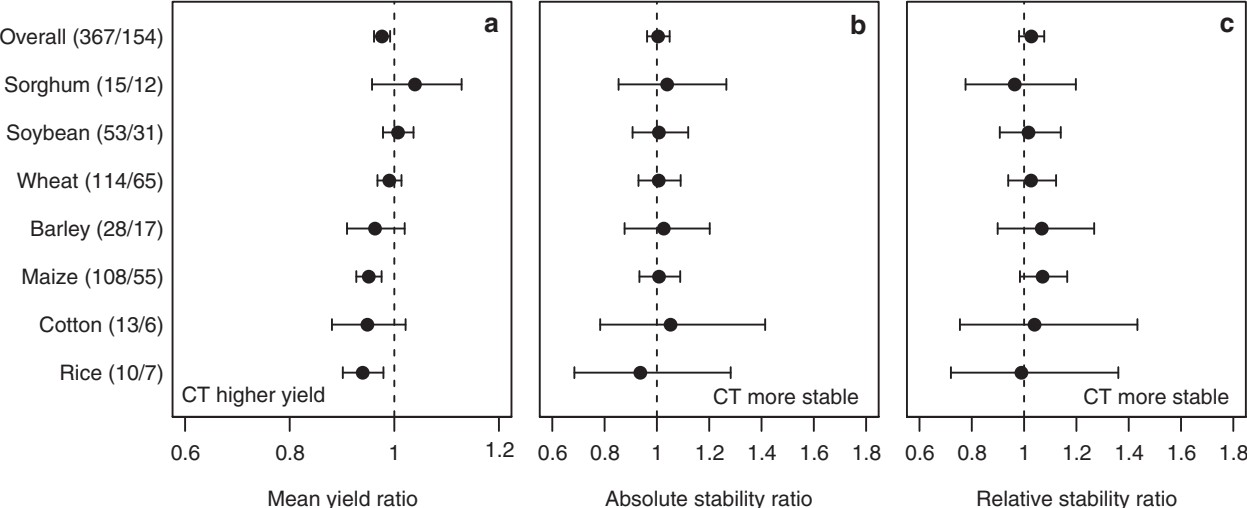

**Fig. 4** Yield and yield stability comparing no-tillage and conventional tillage. **a** Mean yield ratio. **b** Absolute stability ratio. **c** Relative stability ratio of no-tillage (NT) vs. conventional tillage (CT) for all crops (Overall) and for crops, for which at least 10 observations were available. Numbers in parentheses denote the number of observations and studies. A ratio of 1 means that there is no difference between no-tillage and conventional tillage while a value <1 indicates higher yield for conventional tillage. For both stability measures ratios >1 indicate greater stability for conventional tillage. Values are mean effect sizes with 95% confidential intervals. Mean yield or stability were deemed significantly different between no-tillage and conventional tillage if the 95% confidential intervals of the ratios did not overlap one

16% [−10% to −22%] (Fig. 3). Interestingly, our analysis also indicates that the level of P fertilisation influenced, in a similar way to N, differences in yield and yield stability between organic and conventional agriculture (Supplementary Fig. 1). Our analysis further indicated that the addition of green manure had a positive impact on yield and the relative yield stability of organic agriculture (Supplementary Fig. 2).

**Yield stability of conservation and conventional agriculture.** We used the dataset of Pittelkow et al.[12] to compare temporal yield stability of conservation agriculture (focusing on no-tillage) and conventional agriculture. Our analysis indicated that both absolute and relative yield stability did not differ between no-tilled and conventionally tilled fields for the overall dataset and for crop species with at least 10 observations (Fig. 4, see Supplementary Table 2 for all species contained in the dataset).

We then tested whether the application of crop rotation and residue management, two of the main conservation agriculture principles, influenced yield stability. The application of crop

rotation and residue management in no-tillage had, compared to conventional tillage, no effect on absolute and relative stability (Fig. 5). However, without crop rotation and residue management, no-tillage had a 23% [−1% to −50%] reduced relative stability compared to conventional tillage. This result has to be interpreted carefully, as the group where none of the principles of conservation agriculture was followed, was only based on 15 observations (11 studies).

We further tested whether effects of no-tillage and conventional tillage on yield stability depended on climate conditions, comparing dry and humid climate. There was no difference in absolute stability between dry and humid climate and also no difference in relative stability in dry conditions. In contrast, in humid climates, conventionally tilled fields had higher relative yield stability (Supplementary Fig. 3).

## Discussion
Our work adds a new perspective to earlier meta-analyses[12,21] and reveals the effects of different cropping systems on the

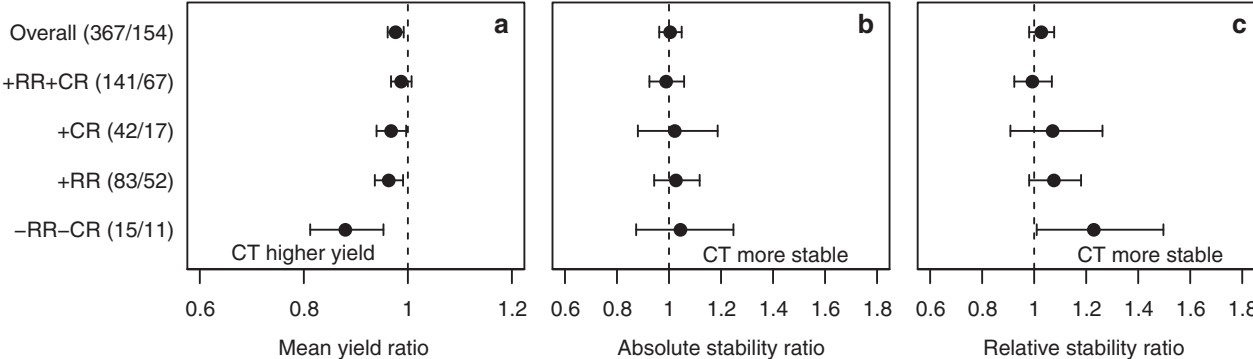

**Fig. 5** Effect of crop rotation and residue retention on yield and yield stability comparing no-tillage and conventional tillage. **a** Mean yield ratio. **b** Absolute stability ratio. **c** Relative stability ratio of no-tillage (NT) vs. conventional tillage (CT) for subcategories of observations regarding residue retention (RR) and crop rotation (CR):+RR+CR (residue retention and crop rotation), +RR (only residue retention), +CR (only crop rotation), or –RR–CR (without residue retention or crop rotation). Numbers in parentheses denote the number of observations and studies. A ratio of 1 means that there is no difference between no-tillage and conventional tillage while values <1 indicate higher yield for conventional tillage. For both stability measures values >1 indicate greater stability for conventional tillage. Values are mean effect sizes with 95% confidential intervals. Mean yield or stability were deemed significantly different between no-tillage and conventional tillage if the 95% confidential intervals of the ratios did not overlap one

variability and reliability of food production across years (e.g., temporal yield stability). Our analysis demonstrated that conventional agriculture has, on average, and per unit food produced, a higher relative yield stability compared to organic agriculture. Yield stability depended on crop species and nutrient management. Notably, the absolute stability of crop yield was the same in organic and conventional agriculture. However, relative stability, which is the temporal variation per unit yield produced, was significantly higher under organic agriculture due to reduced yields in organic agriculture. Thus, per unit food produced, there is higher temporal variation in yield in organic agriculture.

Enhanced fertilisation and the application of green manure were identified as tools to reduce the yield stability gap of organic agriculture with conventional agriculture (Fig. 3; Supplementary Figs. 1 and 2). The observation that fertilisation enhances yield stability is in agreement with Deguines et al.[24] observing that relative yield stability increased with increasing land use intensity. Further experiments need to test whether enhanced fertilisation can reduce the yield gap and enhance yield stability under organic farming. Recommendations for enhanced fertilisation would rely on the assumption that sufficient organic fertilisers are available (e.g., see Muller et al.[25], but see Connor[26]). Moreover, it is important to note that increased fertilisation may raise additional environmental concerns, including the loss of nutrients through leaching and subsequently, enhanced levels of nitrate in drinking water or enhanced production of the greenhouse gas $N_2O$[27]. The positive effect of green manure on yield stability is in agreement with a recent study that showed that green manure (e.g. cover crops) are especially suitable to enhance yields in less intensive cropping systems, such as organic agriculture[28].

The reasons for reduced relative yield stability under organic farming can be manifold and include, beside fertilisation level, enhanced disease pressure (and fewer opportunities to rapidly control pests with pesticides). Also, the timing of fertilisation influences plant yield, and appropriate timing is more difficult with organic fertilisers because nutrient release is delayed compared to readily available mineral fertilisers in conventional agriculture. Moreover, past and current breeding programs have largely focused on high-yielding varieties adapted to work well with conventional inputs[21] and there has been little selection for traits being important in organic agriculture (e.g. increased disease resistance, enhanced cooperation with plant symbionts, better weed suppressing abilities or higher resistance and competitive ability against weeds).

Compared to conventional agriculture, organic agriculture generally has a positive effect on a range of environmental factors, including above and belowground biodiversity[8,29–31], soil carbon stocks[32] and soil quality[10]. Moreover, organic farming can reduce soil erosion[33] and has a reduced global warming potential[34]. However, higher productivity and increased relative stability in conventional agriculture are strengths compared to organic agriculture. Thus, in order to benefit from the strengths of organic farming (e.g., reduced environmental impact and enhanced biodiversity) a multifaceted strategy is necessary to improve its yield and relative yield stability. Such a strategy should focus on enhanced plant nutrition (see above), breeding, weed and disease control, and consider the use of state of the art technologies including precision farming, remote sensing (e.g., through drones or satellites) to detect disease or nutrient deficiency, and robotics (e.g., for weed control)[35]. Moreover, measures such as the inclusion of cover crops (see above) or active stimulation of soil life through soil ecological engineering are especially promising for lower intensity systems such as organic agriculture, and this can further help to reduce the yield gap and the yield stability gap between organic and conventional systems[11,28]. Further studies also need to assess how environmental stresses, such as drought or the negative effects of climate change, influence yield stability in organic and conventional production systems. Finally, when comparing organic and conventional agriculture, it is important to provide an 'output and input footprint' and assess the overall impact of organic and conventional farming practices, including yield, yield stability, energy use, pesticide use, fertiliser use, and overall environmental performance.

Absolute and relative yield stability on average did not vary between no-tillage and conventional tillage indicating that a transition to no-tillage generally does not affect yield stability. Interestingly however, yield and yield stability were affected by climate and no-tillage systems in humid climate had a reduced yield and yield stability compared to dry climate. These differences are probably due to better soil water retention and slightly higher yields of no-tilled soils in dry climate vs. the negative effects of delayed soil warming, nutrient mineralization and reduced soil aeration in no-tilled, wet and heavy soils[7,36]. Note that selection and breeding of crops varieties for conservation agriculture is not yet widespread[37]. Hence, further breeding efforts may enhance yield and yield stability in conservation agriculture.

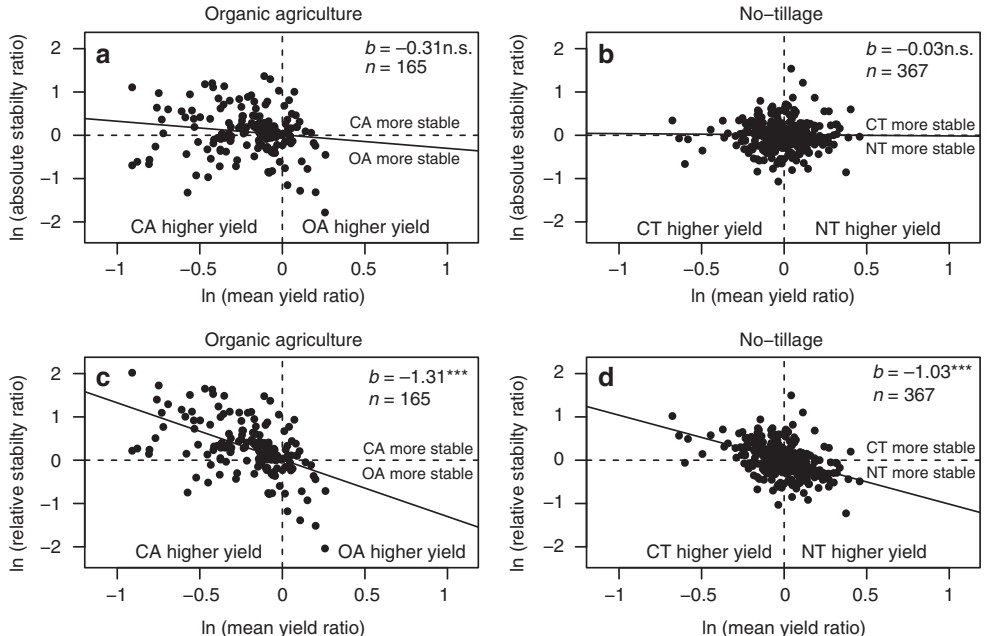

**Fig. 6** Relationship between mean yield ratio and stability ratios. **a**, **b** Relationship of the mean yield ratio to the absolute stability ratio. **c**, **d** Relationship of the mean yield ratio to the relative stability ratio for the dataset comparing organic (OA) and conventional agriculture (CA) (**a**, **c**) and the dataset comparing no-tillage (NT) and conventional tillage (CT) (**b**, **d**), respectively. Each dot represents one multiple year observation (MYO) and ratios are on the natural log scale. The regression line was fitted on log-transformed values, i.e. $\log(y) = a + b \times \log(x)$, where $y$ was the respective stability ratio and $x$ was the mean yield ratio. ***denote significance at $P < 0.001$ for a $t$-test with H0: $b = 0$, and n.s. denotes non-significant ($P > 0.05$)

In our analysis, we employed two different stability measures: absolute stability (measured by the standard deviation in yield across the investigated years) and relative stability, which corrects for yield (measured by the coefficient of variation). While there was a significant difference for relative stability between organic and conventional agriculture, there was no significant difference for absolute stability (Fig. 1). This was also indicated by the negative relationship between the mean yield ratio and relative stability ratio (meaning that relative yield stability increased with increasing yield) (Fig. 6). Hence, the reduced relative stability in organic agriculture is most likely related to reduced mean yield. The absence of a correlation between the absolute stability ratio and the mean yield ratio in the dataset suggests that absolute stability is less affected by yield level. A similar negative relationship between the coefficient of variation and mean yield has been shown previously by Döring et al.[38]. They associated this with Taylor's power law[39], which predicts that the natural logarithm of the variance is proportional to the natural logarithm of the mean. This can lead to a spurious negative relationship of the coefficient of variation and the mean. We therefore investigated the relationship between both stability measures and mean yield, and found that in both datasets absolute stability is not related to the mean yield and relative stability is inversely related to the mean yield (see Supplementary Note 1, Supplementary Figs. 4 and 5). The coefficient of variation has been used extensively to quantify stability[16,17,40,41], but its relationship to the mean yield has rarely been investigated[38]. In light of this, we stress the importance—also for future studies—of distinguishing between relative and absolute stability and, in particular, comparing the relationship to the mean when interpreting results.

The estimated yield gap between organic and conventional agriculture in this study (16%) was slightly smaller than the 19% estimated by Ponisio et al.[21]. This is because we only used 41% of the observations (and 34% of the studies). In our analysis we only included comparisons with a minimum of 4 years of observation per crop (see Methods) explaining this lower number. This

approach was necessary in order to be able to calculate the year-to-year temporal variation, which is necessary for a robust assessment of yield stability. Similarly, Pittelkow et al.[12] demonstrated that, on average, no-tillage reduced yield by 5.7% compared to conventionally tilled fields, while we only observed a difference of 2% [−1% to −4%] (Fig. 4) using 45% of the observations (and 25% of the studies) used in the original analysis. The advantage of our approach is that short-term studies are removed. This reduces the effect of extreme outlier years and generally provides a more robust analysis of differences between these cropping systems. Moreover, this approach also reduces potential transition effects of previous management (e.g. plant yield levels of organic fields that had previously been managed conventionally might be higher because such fields generally still contain enhanced nutrient levels).

For our meta-analysis, we used a different model approach compared to Pittelkow et al.[12] and this may further explain some of the observed differences with that study. Pittelkow et al.[12] applied a weighted mean calculation with bootstrapping, which does not account for the nested structure in the dataset, and leads to non-independence of observations. We corrected for the nested structure of observations derived from the same study by adding a random study effect and by combining observations of several years into multiple year observations. Note, that the datasets used for this study are still based on relatively short-term experiments, i.e. observations with a duration of 4 or 5 years represent 60% of all observations in the dataset for no-tillage and 39% for organic agriculture (Supplementary Fig. 6), pointing to the need for long-term experiments.

It is important to mention that our meta-analysis uses data from diverse systems, geographic areas, and crop species. For instance, the reduced relative yield stability of organically managed fields provides an average response. Studies that aim to enhance yield stability or reduce the yield gap for organic agriculture should evaluate those experiments and conditions where yield or yield stability are higher (or not lower) under organic

agriculture[40] and investigate the causes (e.g. soil type, field management, land use intensity, crop varieties, etc.). Similarly, it is important to investigate under which conditions no-tillage has the most beneficial effects on yield and yield stability.

Our analysis is based on field-scale measurements, and it did not assess yield stability at the farm scale (with a range of crops planted in different fields) or at a regional, national or global scale. To enhance the overall farm level yield stability, farmers could cultivate different crops in different fields (e.g. this reduces the impact of poorly performing crop species at one particular field). Another important strategy to achieve increased yield stability is to grow mixtures of crop species or mixtures of genotypes to exploit positive interaction effects and thus reduce the risk of crop failure[42,43]. Further modelling and work at different scales (e.g., farm, regional, national and global) is necessary to understand how farmers and policy makers can enhance the stability of the food supply. For instance, farm specialisation and the growing of a few crops may lead to increased regional synchrony, increasing the risk of regional crop failures because of climate or pest/disease outbreaks (examples are e.g., wheat yield losses in Australia (2006) or Russia (2010)). Beside temporal yield stability, there are other measures to evaluate management systems, such as, the resilience of different farming practices to disturbance or climate change or the ability of a particular system to produce enough food or income.

Overall, this work provides further information about the performance of organic and conservation agriculture. The assessment of yield stability and the resilience of cropping systems to environmental variability should receive increased attention because reliable agricultural production is a key issue in light of the growing world population and enhanced demands for food. Moreover, climate change and the predicted increase of extreme weather events will provide additional challenges for stable food production.

## Methods
**Data generation**. We used two datasets: (1) a dataset on organic farming by Ponisio et al.[21] comparing the yields of organic and conventional farming and (2) a dataset on no-tillage by Pittelkow et al.[12] comparing the yields of no-tillage and conventional tillage. Both datasets were generated for meta-analysis studies, comprising data from published experiments, and were published as supplemental material. Only field experiments containing side-by-side yield comparisons were included in the database to ensure comparability of the cropping system treatments. Because the focus of this study was on temporal yield stability, i.e., annual variability across years, single year comparisons from the original datasets were combined in order to create observations that were based on several years for each crop investigated (i.e., multiple year observations (MYO)). We focused on studies with a minimum of 4 years of observation for the same crop, thus excluding short-term studies.

**Dataset on organic agriculture**. The original dataset from Ponisio et al.[21] was modified in order to calculate temporal yield stability across years. In order to do this we performed the following steps: First, we corrected a number of minor errors in the original dataset (Supplementary Table 3). Second, we removed all comparisons where the years of observations were not the same for organic and conventional farming. Third, in order to calculate the standard deviation across years, multiple year observations (MYO) had to be compiled: Comparisons from the same experiment that originated from single years were combined into MYOs (for examples see Supplementary Table 4, Supplementary Fig. 7), and comparisons where the collected error term was the variance across years were used as they were. Fourth, MYOs that contained more than one observation from the same year were removed. Fifth, comparisons based on units which could not be transformed to tonnes ha$^{-1}$ (i.e., lb plant$^{-1}$, boxes ha$^{-1}$, kg plant$^{-1}$, bales ha$^{-1}$, trays ha$^{-1}$, bales ha$^{-1}$, kg (square centimetre of limb cross-sectional area)$^{-1}$, ka ha$^{-1}$, g, kg tree$^{-1}$, kg Fw plant$^{-1}$) were removed (this affected a total of 19 comparisons). Sixth, in order to have a robust estimate of the temporal yield stability, we required a minimum of 4 years of observation for each MYO and thus all comparisons based on <4 observations were removed. Finally, when investigating the standardized residuals one extreme outlier was detected and was removed to achieve normal distribution of residuals (Supplementary Fig. 8).

After these steps, the final dataset on organic agriculture contained 165 multiple year observations from 39 studies that were based on 443 comparisons from the original dataset.

**Dataset on no-tillage**. The original dataset from Pittelkow et al.[12] was processed in the same way as mentioned above for the dataset of Ponisio et al.[21] with the following additions: Comparisons containing a zero (i.e., no yield data available) for either no-tillage or conventional tillage were removed. As we used the duration value as year of observation, all comparisons containing NA in the 'study duration' column were removed. Similarly to the approach for the dataset by Ponisio et al.[21], MYOs were created by combining comparisons of different years from the same experiments. However, when creating the MYOs, we observed that for some comparisons the number of rows per MYO was greater than the duration length (see column subtreatments in Supplementary Table 5). After we compared these observations with the original publications, it was clear that these MYOs were derived from different subtreatments. Single observations used for creating MYOs were thus either collected in subsequent or alternating order (see column order in Supplementary Table 5). In order to separate subtreatments within MYOs the number of rows needed to be in agreement with the duration of the study, and all MYOs that did not fulfil these criteria were removed. The remaining MYOs containing subtreatments were then further split into separate MYOs (column MYO in Supplementary Table 5). Similarly to the dataset on organic agriculture, standardized residuals were investigated and two extreme outliers were removed (Supplementary Fig. 9).

In the end, the final dataset on conservation tillage contained 367 multiple year observations from 154 studies that were based on 2453 comparisons from the original dataset.

**Statistical analysis**. After the creation of the multiple year observations, for each MYO the mean yield ($X$), standard deviation (SD) and number of years of observation ($N$) was available for the experimental (e) and the control (c) treatment. In the dataset on organic farming, the organic treatment was used as the experimental treatment, and the conventional treatment was used as the control treatment. In the dataset on no-tillage, the no-tillage treatment was used as the experimental treatment, and the conventional tillage treatment was used as the control treatment.

In order to determine the overall difference in mean yield we used the log response ratio (expressed as mean yield ratio) as effect size, which is the natural log of the ratio of the mean yield of both cropping systems[44]. The log-transformation has the property to produce normally distributed data[45]. Following Nakagawa et al.[46] we used the two following measures to asses temporal stability: (1) the 'absolute stability ratio', which is based on the standard deviation of both treatments as an indicator for variability, and (2) the 'relative stability ratio', which is based on the coefficient of variation (CV: standard deviation across years divided by the mean across those years) of both treatments as indicator for variability. Therefore, in the latter measure the variability is standardized per unit yield (i.e., the variability relative to the yield level).

For each of the three measures, the ratio was calculated by dividing the respective response of the experimental treatment (organic farming or no-tillage) by the respective response of the control treatment (conventional farming or tillage, respectively). A ratio greater than one indicates greater yield or greater variability (i.e., reduced stability), respectively, for the experimental treatment. The equations for the respective responses were:

$$\ln(\text{mean yield ratio}) = \ln\left(\frac{X_e}{X_c}\right), \tag{1}$$

$$\ln(\text{absolute stability ratio}) = \ln\left(\frac{SD_e}{SD_c}\right) + \frac{1}{2(N_e - 1)} - \frac{1}{2(N_c - 1)}, \text{ which simplifies with } N_e = N_c \text{ to}$$

$$\ln(\text{absolute stability ratio}) = \ln\left(\frac{SD_e}{SD_c}\right), \tag{2}$$

$$\ln(\text{relative stability ratio}) = \ln\left(\frac{CV_e}{CV_c}\right) + \frac{1}{2(N_e - 1)} - \frac{1}{2(N_c - 1)}, \text{ which again simplifies to}$$

$$\ln(\text{relative stability ratio}) = \ln\left(\frac{CV_e}{CV_c}\right), \tag{3}$$

with $CV_e = \left(\frac{SD_e}{X_e}\right)$ and $CV_c = \left(\frac{SD_c}{X_c}\right)$.

In order to account for the sampling uncertainty in each observation we used the sampling variances as proposed in Nakagawa et al.[46]. Through the inclusion of the sampling variance observations with better sampling quality (lower sampling variance) receive a greater weight in the analysis. Following Nakagawa et al.[46] the equations for the sampling variances for three different response ratios were as

follows:

$$\mathrm{var}(\ln(\text{mean yield ratio})) = \frac{SD_e^2}{N_e X_e^2} + \frac{SD_c^2}{N_c X_c^2}, \qquad (4)$$

$$\mathrm{var}(\ln(\text{absolute stability ratio})) = \frac{1}{2(N_e-1)} + \frac{1}{2(N_c-1)}, \qquad (5)$$

$$\mathrm{var}(\ln(\text{relative stability ratio})) = \frac{SD_e^2}{N_e X_e^2} + \frac{1}{2(N_e-1)} + \frac{SD_c^2}{N_c X_c^2} + \frac{1}{2(N_c-1)}. \qquad (6)$$

Note that we modified the equation for the sampling variance of the relative stability ratio, because for normally distributed data the variance and the mean are not correlated. Calculations were performed as implemented in the metaphor package.

As some observations shared common control or experimental treatments, we employed a 0. (VC) matrix to correct for correlations among observations following Lajeunesse[47]. When multiple treatments share a common control or common experimental treatment, the assumption of independence is violated. Thus, the effects should be aggregated by using an appropriate variance–covariance matrix. When all observations are independent, this variance–covariance matrix only holds the variance on the diagonal. Following Lajeunesse[47], for two experimental treatments A and B, which have both been compared to the same control treatment C, the variance–covariance matrix holds the variance of the comparisons of A to C (resp. B, C) on the diagonal and the variance of the log of the mean of the control treatment $(\mathrm{var}(\ln(X_c)) = \frac{SD_c^2}{N_c X_c^2})$ on the off-diagonal:

$$\mathrm{VC}(\ln(\text{mean yield ratio})) = \begin{bmatrix} \frac{SD_c^2}{N_c X_c^2} + \frac{SD_A^2}{N_A X_A^2} & \frac{SD_c^2}{N_c X_c^2} \\ \frac{SD_c^2}{N_c X_c^2} & \frac{SD_c^2}{N_c X_c^2} + \frac{SD_B^2}{N_B X_B^2} \end{bmatrix}. \qquad (7)$$

For the responses absolute stability ratio and relative stability ratio, the respective sampling variance–covariance matrices are then:

$$\mathrm{VC}(\ln(\text{absolute stability ratio})) = \begin{bmatrix} \frac{1}{2(N_c-1)} + \frac{1}{2(N_A-1)} & \frac{1}{2(N_c-1)} \\ \frac{1}{2(N_c-1)} & \frac{1}{2(N_c-1)} + \frac{1}{2(N_B-1)} \end{bmatrix}. \qquad (8)$$

$$\mathrm{VC}(\ln(\text{relative stability ratio})) =$$

$$\begin{bmatrix} \frac{SD_c^2}{N_c X_c^2} + \frac{1}{2(N_c-1)} + \frac{SD_A^2}{N_A X_A^2} + \frac{1}{2(N_A-1)} & \frac{SD_c^2}{N_c X_c^2} + \frac{1}{2(N_c-1)} \\ \frac{SD_c^2}{N_c X_c^2} + \frac{1}{2(N_c-1)} & \frac{SD_c^2}{N_c X_c^2} + \frac{1}{2(N_c-1)} + \frac{SD_B^2}{N_B X_B^2} + \frac{1}{2(N_B-1)} \end{bmatrix} \qquad (9)$$

For the generation of the variance–covariance matrix we used a modified version of the covariance_commonControl() function from the metagear package[48]. When testing the effect of moderators (see below), the structure of common control or experimental treatments changed because observations within studies were derived from different levels of the moderator variable. Therefore, a new variance–covariance matrix was created for each moderator.

We employed a mixed model approach using the rma.mv() function from the metafor package in R[49] with REML estimation. To account for variation between studies, a random effect for study was included, and the respective sampling variances (as described above) were included. To estimate the overall effect, a mixed model containing only a fixed intercept and the random study effect was run.

Both datasets contained additional explanatory variables (e.g., crop species or information on management practices such as fertilisation level or the use of green manure). These explanatory variables (moderators) were tested with a separate model for each variable, in which the variable was included as a categorical, fixed effect variable. In order to get the average estimates of the factor levels of the moderator variables, a model was fitted without the intercept. For both, the overall effect and average estimates of the factor levels, 95% confidence intervals, as provided by the rma.mv() function for the coefficients, were used to test the significant difference from 1. All calculations were done with the R statistical package[50].

## Data availability
The original dataset by Ponisio et al.[21] was downloaded from the Dryad Digital Repository (https://doi.org/10.5061/dryad.hf305) and the original dataset by Pittelkow et al.[12] was obtained from supplemental material on the journal's homepage. The datasets and the R-script for the mixed-model analysis generated during the current study are available in the Figshare repository (https://doi.org/10.6084/m9.figshare.6743798).

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

## Acknowledgements

We would like to thank Fabian Scheipl for mathematical support and Raphael Wittwer, Kyle Hartman and Thomas Döring, for constructive and helpful discussion and comments on the manuscript. Funding was provided by Agroscope and the Swiss National Science Foundation (grant number 166079).

## Author contributions

S.K. and M.v.G.A.d.H. conceived and designed the study. S.K. performed the analysis. S. K. and M.v.G.A.d.H. wrote the manuscript

## Additional information

**Competing interests:** The authors declare no competing interests.

