## [Peer Review File · Nature Communications]

Reviewers' comments:

Reviewer #1 (Remarks to the Author):

Comments for authors

This is an interesting and novel work, that may attract considerable attention, and also some controversy. Hence it is important to get things right, which is why I am quite critical in several respects, with the hope of being constructive.

There are several issues that must be addressed and discussed:

- Problems associated with the measure of variation (CV)
- Problems with extrapolating from single crops on single plots/fields to farms and food security in general
- The problem that the increased field scale yield stability (if real) in CA is actually bought at the expense of the environment or increased pressure on resources elsewhere, i.e. agricultural inputs like energy, nutrients or land.

1. Problems with using coefficient of variation (CV) as the measure of stability.

Although using CV seems common and straightforward, when actually using CV there are a number of issues that need consideration. I have mainly done this in the context of population variability, but the issues are very similar when comparing yield variation (or conversely stability). Some key early references are given below.

Firstly, CV is only appropriate as a measure of variation if SD (S) is proportional to the mean (which I will denote X). For populations (and yields are in fact populations), Krebs (Ecological Methodology, 1999) emphasizes that it is only appropriate when the slope of Taylor's Power Law is equal to 2 (i.e. the standard deviation is proportional to the mean). It is clear from the results here that this is not the case here, rather S is independent of X in the data. The authors also note this (lines 135-137) but don't see the problem. It is also evident in Figures S2, S3 and S4, these (or at least some of them) should be included in the main text. The consequence of absolute variation (S) not differing between organic (OA) and conventional (CA), but relative variation being different is noted, but it should be given more attention.

Secondly, McArdle et al. (J anim ecol 1990) discusses measuring variability (in the context of population variability), pointing out among other things that (1) temporal variability depends not only on the spatial and temporal scales of the sampling unit but also on the number of samples taken, that is on the duration of the study; the longer the sampling period the greater the variability. This is because environmental variation increases as the temporal scale increases (it has a "red spectrum" or "red shift"; Pimm & Redfearn Nature 1988). This means that longer time series are likely to show more variation (higher CV), and hence comparisons should take this into account. This is not done appropriately in the MS, rather the increased weight given to longer time series (presumably more precise) increases this effect, which then means confounding the weighing procedure with the environmental variation effect.

McArdle et al and others note that the usual measure of the coefficient of variation ($CV=S/X$) is slightly biased, and an approximate bias corrected estimator is available: $(1 + 1/4n) (S.D./X)$; where n is the sample size (Haldane 1955). This is important for small sample sizes (as in this study), and this is therefore the adjustment for duration length that should be used, not the unfounded one used (see below). Note however, that this correction depends on whether the data are normally or log-

normally distributed, so the authors need to check their distributions of the data for OA and CA (NT vs T) before deciding on the adjustments. See the Wikipedia on CV, for example.

In addition, McArdle and others argue that because of sampling errors, any statement of variability must be accompanied by some measure of the confidence in the estimate of variation. This is not done in the MS. Gaston & McArdle (PRSB 1994) argue that, many, if not most, reported measures of variability actually have such large confidence intervals as to make sensible comparisons virtually impossible. The authors have not estimated any measures of precision, but my (unfounded perhaps) feeling is that this would make much of the results extremely imprecise. The studies should at least have been weighed based on the precision of the estimates, but this was not done (see also below on replicates in the OA-CA comparison).

Hence, while the idea of the paper is very clever, the use of CV as the measure of stability is problematic. There are statistical reasons for this, but also agronomic and biological. What is it actually that influences CV of crop yield? Since there is an upper limit on crop yields (which varies according to soil, climate, etc.) variation will be constrained as yields approach the capacity of the soil and environment to support crop production. Is this the explanation for the results? - which suggests that as resource inputs increase to approach that limit, these external inputs (fertilisers, pesticides, etc.) necessarily increases yield stability (see below). Another reason may be lack of breeding for OA (as discussed on lines 153-56). Measuring yield stability is (perhaps) OK, but the mechanisms behind are important to disentangle.

2. Yield stability for single crops at the field level is not necessarily relevant for farm economy or food security.

The authors state a number of times (lines 19, 28, 31 in the Abstract, lines 69, 71) that yield stability (for crops on the paired plots/fields in the data set) translates into food supply predictability, reliable food production, or even global food security. However, this scaling up is in my view not allowed without much more complicated assessments. First, the four crops may be bulk crops, but not the necessarily the ones that confer food security for many people and farmers globally. The food system does not work like that. So, what is to be stabilised – individual crop yields in a variable world, or food provision? And at which scale?

On the farm level, one of the main advantages of using crop rotations is to counter single-crop variability effects on farm income or food production. Crops vary, often asynchronously, in their responses to the environment – weather, pests, etc. – and to economic price volatility, and at the farm level it is the aggregate result on farm economy of the DIFFERENT crops grown that impacts farm performance and food production. If wheat fails, or maize, then beans or vegetables may make up for it. Field scale measures like the ones in this MS are only appropriate at the field scale, but not for farm economy unless farmers are indeed using only one or two crops. But real OA and CA farms usually grow more crops, and actually crop diversity is often higher on organic farms. Therefore the field-scale measures are not proper for comparisons at the farm level. At the farm level, diversification is likely to confer stability, but this is not addressed at all.

On a regional, national or global level, the simple scaling up is even less appropriate, for both agronomic/ecological and economic reasons. This scaling up requires a completely different approach, that utilises field, farm and regional data in novel ways. For example, fewer crops (in CA) and farm specialisation may entail increased regional synchrony, increasing the risk of regional crop failures because of regional climate or pest/disease outbreaks (examples are e.g. wheat in Australia 2006, 2009, Russia 2010, etc). This is something that needs to be included in such scaling-up assessments. Diversification would be a means to counter such effects at all scales, but the present analysis is not

useful for these questions, not even as a background.

Hence the authors should delete all simplified references to food security, only acknowledge that analyses of farm, national and global food security and farm economy need field-scale measures to understand how farmers and policies can enhance stability of food (not the four crops) provision. They should clearly state that this analysis is for plot/field scale paired comparisons. And nothing else.

3. Yield stability in highly productive systems is bought with increased external inputs and at the expense of the environment.

Assuming that CV does measure yield stability, it would be relevant to ask what it is in CA that makes yields more stable than OA. The authors touch on this several times but not enough, in my view. Conventional farming uses more inputs than OA, from energy (N-fertilisers, machinery) and fertilisers (e.g. P, K, micronutrients), to pesticides, herbicides, fungicides, etc. (but OA is not innocent as regards machinery and various types of organic manure). So the "input footprint" is likely larger for CA. Several recent meta-analyses and reviews (e.g. Seufert et al. 2017, Reganold & Wachter 2016, Tuck et al. 2014) show that OA - on average - performs better when it comes to environmental impacts and biodiversity, although exceptions can be found and N-leaching remains a problem for all farming. This means that the "output footprint" may also be larger for CA. Inputs in CA thus have the effect of insuring yields, i.e. increasing yield stability, at the expense of resource use elsewhere. Were there any estimates of nutrient leaching or other input-outputs in the studies used?

The consequence of this for sustainability is still debated, but the authors hardly touch on these issues. Is the increased field level single crop stability in developed countries really sustainable, or is its lack of sustainability bought at the expense of excluding today's resources from developing countries today and tomorrow's food production globally? Is it causing large environmental effects, like nutrient loss, or biodiversity loss? Is it causing effects far away in distant countries, like water eutrophication, climate change or land use changes? CA seems to use less local resources, which means a larger footprint globally. Which systems are likely to be the most resilient to environmental, economic or social perturbations? A balanced discussion on the environmental effects of managing intensified agriculture for yield stability is needed - not long, but addressing the larger-scale implications of the increased yield stability that managing for yield quantity seems to lead to.

Comments on the text:

Lines 73-81. A more in-depth discussion on CV is needed (see above). In the present study, SD (S) does not vary with mean yield (X) so the assumption that CV compares different levels of variation is not really fulfilled. For biological data, it is not clear if the assumption should hold, in fact S (and hence CV) can vary in different ways with X, and this should be examined in more detail before the measure is used (note also the corrected CV formula above).

Lines 89-92. The measure of yields on fields does not account for the fact that OA can counter field and crop yield variability on the farm level. This study is only relevant for field level assessments, but at farms and landscapes it is only relevant if climate, weeds and pest outbreaks are synchronous over larger scales AND this is not countered by crop diversification and crop rotations. So the authors should think a bit more about the possibility to scale up.

Line 96. Sentence should be "yield stability in fields of single crops of CA was ... higher".

Line 105-107. This statement is unfounded. The confidence intervals for yield stability for the 4 crops highly overlap, so there is NO SIGNIFICANT difference between maize and the other crops in Fig. 1.

The only thing that can be said is that maize was not different from 1, but the other crops showed lower stability in OA. The authors need to check this all over the MS, differences between crops and treatments are examined by looking for overlaps among the crop/treatments confidence intervals, not by looking at whether the CI:s differ from 1 or not. This is basic statistical knowledge, so please check carefully.

Line 126-27. The test of fertilisation regime is good! But the increase in stability with intensification does indeed suggest that stability is bought with external inputs, i.e. there are environmental and societal COSTS of increasing yield stability.

Line 134-144. It is good that it is noted that differences in CV are mainly due to differences in mean yield, but the authors should consider its implications. Also, is this effect visible also in CA only or OA only – i.e. plots of CV against X (or logCV vs logX) should be negative also within farming systems. Is this the case? If so, this suggests something interesting and potentially general about increasing yields and yield stability. If not, it would be interesting to understand why.

Line 145. Sentence should read "cause for increasing field scale yield stability" ...

Line 153-57. The point on the lack of breeding is important and should be kept.

Lines 159-196. I wonder if these results, which are interesting, can be scaled up to farm and regional scales. In any case, the analysis suffers from the same problems as discussed above.

Line 181-82. See comment concerning lines 105-107. Maize does not differ from the other crops! Correct in next version!

Line 207. Add that breeding for resistance or competitive ability vs. weeds may also be needed, as dicussed above.

Line 234- ... Conclusions. May need changes after the problems above have been addressed.

Materials and Methods

Line 255. One could argue that because the systems differ in inputs, the treatments are in fact not comparable. Any kind of comparison of OA-CA or NT-T entails comparing systems that differ in inputs. Needs a short discussion-

Line 263-64. This is sensible, but I lack a discussion of how the small sample sizes affect the precision of the CV estimates, how this can be estimated and how it should be included in the analysis. Note also that environmental variation increases with duration, which should (could) be accounted for with e.g. analysis of residuals from a yield stability against time regression, rather than just the CV or S. However, since the data seems to be paired, this is only important of these relations really differ between CA and OA, or NT and T.

Line 294-95. Does this mean that data were indeed missing and that it was not a measured 0 (zero) because of crop failure?

Line 308-310. Give the exact formula used, please.

Line 311-317. This is a most confusing paragraph. In the Pittelkow et al. paper, I think they were only examining mean yields, so this procedure weighed longer studies higher because of their higher

precision. This COULD make sense for yield stability too, but since there are other issues like the increased environmental variation with time, the rationale behind this weighing becomes obscure to me. Why not examine the relations of yield stability vs. duration first, so see if there is any relation that is negative (indicating environmental variation effects) and then re-consider what the weighting should be. You need to think about this very carefully, because weighing longer studies more implies that you confound environmental variation with precision, and thus give more weight to studies with larger environmental variability. Is this what you want?

Line 322-324. Does this really mean that plot-n=1 for most of the OA-CA comparisons?? This just can't be the case!! It would mean that MOST comparisons are UNREPLICATED, which is such a dismal finding that it should be highlighted. Or did you just not bother to look for the within-study replication?

Figure 1. Barley differs in yield ratio from Maize and Soybean, but there is no crop difference in stability ratio. You should also add the SD comparison here.

Fig. 2. More N and similar N differ in yield ratio, but no differences with N input in stability ratio.

Supplementary materials:

Fig S2. (c) OA is NOT more variable in 1/SD since the CI:s overlap 1 in all cases.

Fig S3 (c) Again, OA is NOT more variable in 1/SD.

Fig S4. It's interesting to see this figure, because it highlights how you may use different measures of stability. As argued in McArdle et al, SD is actually also useful to measure stability – it depends on what you are looking for and the structure of the data. You assume that relative variation is the important thing = CV. But why? In this particular study, you can use CV, S or range from the shown data. You could also look at risk of crop failure (but that's not really happening; risk for economic loss could perhaps also be a relevant measure). In any case, CV varies, but S does not, and range not really (CA \approx 4.5, OA \approx 4.8, hardly significant). What does this mean? That measures are not showing consistent results, and THE CHOICE that YOU make on what is interesting is important. I don't think you make a good argument for your choice of CV on lines 76-80 or 134-140, you just uncritically use it. If you still want to use CV (which is OK) the pros and cons should be clearly discussed (see above).

Fig S7. Explain the +RR and +CR in the legend. The reader will have forgotten them.

Table S2. What does the column "Comparison" mean? Explain.

ALSO: This table should (when/if published) contain all the data used (it is electronic) and include n-values for the treatments too.

General final words:

I liked this paper a lot, despite my critical comments. It highlights the complexities involved in discussions on food and yield stability, security and – of course – food distribution locally, regionally and globally. There are many problems that need the inputs from this (and other) analyses, in order to get these discussion on the right track. Issues that are important are, e.g.:

- Spatial and temporal scales affecting stability and variability.
- Crop distributions and crop rotations; what should be stabilised, individual crops or food production in a variable world?
- The confusion between food security and local crop stability.

- How do we measure yield stability? What are the drivers of variation in yields and yield stability – agronomically, ecologically, economically? Does economy matter for subsistence farmers? Should we minimize the risk of individual crop and total food production failures?
- Can we compare systems that differ in the amount of inputs and their environmental effects, and if so how?

These are interesting areas for future research, so I hope you can address my concerns above so that discussion on these important issues can progress.

Jan.bengtsson@slu.se

Reviewer #2 (Remarks to the Author):

A meta-analysis of yield stability in conventional , organic 1 and conservation agriculture

General comments

The topic of yield stability is been discussed frequently among scientists and therefore the meta-analysis of yield stability in various cropping systems is timely. The authors used a subset of two already published data sets, which were reported/published earlier to compare yield between organic and conventional farming systems and to determine yield differences between no-till (including conservation ag) and tilled systems. These two publically available data sets were used here to calculate yield stability.

My main concern with the analysis is how yield stability was calculated. The key issue is that stability is calculated as the ratio of the mean to variability around that mean. This has the effect of skewing the highest yielding system toward higher stability. One way to look at this is by re-ordering the equation, which reveals that to have equal stability of the two systems, a lower yielding system has to have lower variability around the mean. By this definition, a high yielding but highly variable system can have the same "stability" as a lower but consistent yielding system. The clearest example is Fig. S4. Comparing the two systems, they appear to have roughly the same variability around the mean (conventional might even be slightly more variable). However, the conventional system has higher yield stability merely because it has a higher yield.

Should the real issue here be yield stability or should it be yield resilience? Yield stability is about consistency, but are we concerned if there is a year with an above average yield? A year with higher yield equals to higher variability in yield, and therefore lower stability. Should we try to avoid years with poor yield, which involves resilience to disruptions? High variability will be OK when the outcome is higher yield.

Therefore, should the analysis/manuscript focus on resilience in yield and not on stability in yield as defined here?

A second but less of a concern is the conclusions drawn from the tillage data set. Did Pittelkow et al come to the same conclusion as stated here on lines 173-175? Likewise, did Pittelkow et al. already concluded what is stated on lines 178-181? In general, when using a data set already analyzed/published, the authors who are (re)-using this data set should be careful to report what is a new finding or what is merely stating again what has already been reported in the earlier publications.

Specific comments:

Line 101. Wheat does not show a significant difference in Fig. 1 as it crosses the 0 line.

Line 139: Why is there a reference here to Fig. 1? Are the supporting data only shown in Fig. S2?

Reviewer #3 (Remarks to the Author):

see following page

Heijden and Knapp compare the yield stability of organic, conventional and no-till agriculture of major cereals crops. I have three major concerns:

1) Without reading the methods in detail, the article appears to compare cropping systems broadly, when only four crop species had enough data to be compared. Throughout the article (in the abstract, title, etc.) that this is a comparison of only four crops (though major important crops) should be made clear.

2) The coefficient of variation, when applied to a small or moderately sized sample, tends to be too low. The corrected version should be used:

$$\hat{cv} = (1 + 1/4n) * cv$$

3) The authors follow the analytic methods of Pittelkow et al. These methods do not account for the fact that multiple CV are extracted from the same study. They also do not account from random variation between studies. Like in Seufert et al. this study thus suffers from pseudo-replication which inflates the type I error. The authors should re-analyze this data set using a random effects meta-analytic model:

$$\begin{aligned} y_{ij} &= \mu + \alpha_i + \epsilon_{ij} \\ \alpha_i &\sim N(0, \sigma_\alpha^2) \\ \epsilon_{ij} &\sim N(0, S_{ij}) \end{aligned} \tag{1}$$

where y_{ij} is the observed magnitude of the j^{th} \hat{cv} of the i^{th} study, μ is the mean \hat{cv} across studies, α_i is the effect of i^{th} study, and ϵ_{ij} is the residual. σ_α^2 is the between study variance, and S_{ij} is the variance of \hat{cv}_{ij} . The funny thing here is that the cv is a cv of means, so it is not immediately clear to me what estimate should be used. Perhaps the SE of the CV would be sufficient? $\hat{cv}/4n$. It would be best to come up with a clever way to incorporate the reported SD around the mean yield for each year reported in the studies.

Also, what was done when multiple organic treatments were paired with the same conventional control? (i.e., a multi treatment experiment in the original study). This is not clear from the methods.

After the authors re-analyze the data I would like to read the manuscript again because many results may change.

This is an interesting and novel work, that may attract considerable attention, and also some controversy. Hence it is important to get things right, which is why I am quite critical in several respects, with the hope of being constructive.

There are several issues that must be addressed and discussed:

- Problems associated with the measure of variation (CV)
- Problems with extrapolating from single crops on single plots/fields to farms and food security in general
- The problem that the increased field scale yield stability (if real) in CA is actually bought at the expense of the environment or increased pressure on resources elsewhere, i.e. agricultural inputs like energy, nutrients or land.

Thank you very much for the many helpful comments and suggestions. Below we address each point separately.

1. Problems with using coefficient of variation (CV) as the measure of stability.

Although using CV seems common and straightforward, when actually using CV there are a number of issues that need consideration. I have mainly done this in the context of population variability, but the issues are very similar when comparing yield variation (or conversely stability). Some key early references are given below.

Firstly, CV is only appropriate as a measure of variation if SD (S) is proportional to the mean (which I will denote X). For populations (and yields are in fact populations), Krebs (Ecological Methodology, 1999) emphasizes that it is only appropriate when the slope of Taylor's Power Law is equal to 2 (i.e. the standard deviation is proportional to the mean). It is clear from the results here that this is not the case here, rather S is independent of X in the data. The authors also note this (lines 135-137) but don't see the problem. It is also evident in Figures S2, S3 and S4, these (or at least some of them) should be included in the main text. The consequence of absolute variation (S) not differing between organic (OA) and conventional (CA), but relative variation being different is noted, but it should be given more attention.

RESPONSE: In order to address these useful points, we have made several major changes to the manuscript. First, we calculate and present two measures of yield variability: the absolute stability (which is the standard deviation of yield across years) and the relative stability (which divides the variability across years (expressed as standard deviation) by the mean yield over the same period, i.e. this is the coefficient of variation)(see text and figures). In the general discussion, we stress that it is important to differentiate between these two measures.

Second, we agree that concerns related to Taylor's power law (TPL) are relevant. In order to test the TPL relation, we included a regression of the log(SD) on log(mean) of observed values (not ratios) for each treatment separately (Supplementary Figure 4) and also of treatments within each observation (Supplementary Figure 1). In addition we also included a Supplementary Note on this issue. While SD is correlated to the mean across observations, there is no correlation between treatments within observations. As we only compare treatments within observations by using ratios, we therefore argue that the use of CV is valid. However, as SD is similar between treatments, and the slope of log(CV) on log(mean) is around -1 (and also when regressing ratios), we acknowledge that the difference in relative stability ratio is mainly due to the difference in mean yield.

Secondly, McArdle et al. (J anim ecol 1990) discusses measuring variability (in the context of population variability), pointing out among other things that (1) temporal variability depends not only on the spatial and temporal scales of the sampling unit but also on the number of samples taken, that is on the duration of the study; the longer the sampling period the greater

the variability. This is because environmental variation increases as the temporal scale increases (it has a "red spectrum" or "red shift"; Pimm & Redfearn Nature 1988). This means that longer time series are likely to show more variation (higher CV), and hence comparisons should take this into account. This is not done appropriately in the MS, rather the increased weight given to longer time series (presumably more precise) increases this effect, which then means confounding the weighing procedure with the environmental variation effect.

RESPONSE: Thanks for critically considering this. We now checked if longer observations show higher variation (see Figure 1 below). Besides the higher variation in SD in observations derived from up to 6 years of observation, we could not discover a trend of increasing variability with more years of observation.

Figure 1: Relation of Log(SD) and number of years of observation

McArdle et al. (1990, cited above) is mainly concerned with the autocorrelation between successive sampling periods. While this might be true for e.g. population densities, we argue that this is not the case for yearly variation in crop yields, where variation is highly determined by (random) weather conditions and disease and we believe that there is little reason to assume a higher autocorrelation between successive years than between more distant years. Moreover, in our new analysis we now used a mixed model approach, where we used sampling variances for the weighting, following common meta-analysis practice. Therefore the direct weighting by the number of years of observation is removed.

McArdle et al and others note that the usual measure of the coefficient of variation ($CV=S/X$) is slightly biased, and an approximate bias corrected estimator is available: $(1 + 1/4n)$ ($S.D./X$); where n is the sample size (Haldane 1955). This is important for small sample sizes (as in this study), and this is therefore the adjustment for duration length that should be used, not the unfounded one used (see below). Note however, that this correction depends on whether the data are normally or log-normally distributed, so the authors need to check their distributions of the data for OA and CA (NT vs T) before deciding on the adjustments. See the Wikipedia on CV, for example.

RESPONSE: Thanks for the suggestion of this correction. We were not aware of this. While this correction factor is of particular importance when comparing variation (measured by CV) between treatments or species, that are based on a different (and small) number of observations per treatment or species, in our analysis we always compare the same number of observations (i.e. years) for both treatments. Furthermore, as our analysis is based on pairwise ratios of the CVs of both treatments and each pairwise observation has the same number of years of observation (n) this correction factor cancels out due to:

$$CVR = \frac{CV_{exp} * (1 + \frac{1}{4n_{exp}})}{CV_{cont} * (1 + \frac{1}{4n_{cont}})} \text{ with } n = n_{exp} = n_{cont} \text{ simplifies to } CVR = \frac{CV_{exp} * (1 + \frac{1}{4n})}{CV_{cont} * (1 + \frac{1}{4n})}$$

and thus $CVR = \frac{CV_{exp}}{CV_{cont}}$.

We did therefore not use this correction factor in our new analysis.

In addition, McArdle and others argue that because of sampling errors, any statement of variability must be accompanied by some measure of the confidence in the estimate of variation. This is not done in the MS. Gaston & McArdle (PRSB 1994) argue that, many, if not most, reported measures of variability actually have such large confidence intervals as to make sensible comparisons virtually impossible. The authors have not estimated any measures of precision, but my (unfounded perhaps) feeling is that this would make much of the results extremely imprecise. The studies should at least have been weighed based on the precision of the estimates, but this was not done (see also below on replicates in the OA-CA comparison).

RESPONSE: We now have included confidence intervals for all ratios reported in the text. Also, all plots for yield and stability ratios include confidence intervals. In the new analysis using a mixed model approach, the sampling variance of each observation is weighed through the inclusion of a sampling variance term in the model, following common meta-analysis practice (see Material and Methods).

Hence, while the idea of the paper is very clever, the use of CV as the measure of stability is problematic. There are statistical reasons for this, but also agronomic and biological. What is it actually that influences CV of crop yield? Since there is an upper limit on crop yields (which varies according to soil, climate, etc.) variation will be constrained as yields approach the capacity of the soil and environment to support crop production. Is this the explanation for the results? - which suggests that as resource inputs increase to approach that limit, these external inputs (fertilisers, pesticides, etc.) necessarily increases yield stability (see below). Another reason may be lack of breeding for OA (as discussed on lines 153-56). Measuring yield stability is (perhaps) OK, but the mechanisms behind are important to disentangle.

RESPONSE: We fully agree, that the use of CV as stability measure has certain problematic aspects, and that we did not emphasize this issue sufficiently. We therefore re-analysed the data and we now clearly present the real variability across years (absolute stability) and the variability that is standardize per unit of yield (relative variability). In the text and discussion we treated both measures of stability them with equal importance. In order to test for their relation to the mean yield level, we tested regressions between the stability ratios and mean yield ratio (Figure 3), and variation measures (SD and CV) between observation within treatments (Supplementary Figure 8), and between treatments within observations (Supplementary Figure 9). Also we included a discussion on this issue in the main text and added a Supplementary Note. We would like to point out that the coefficient of variation is widely used and we feel that in combination with a measure for the absolute variability we present the results in a transparent and correct way. We have now further discussed the mechanisms behind the differences in stability (see lines 269-280 for organic farming and 336-352 for no-tillage).

We appreciate the suggestion to further look into the mechanisms explaining the results. Hence, in order to address this we now performed an additional analysis and included explanatory variables in the data-set. In order to address the role of increased resource input, we analysed the effect of N and P fertilization (these data were available in the originals data sets) on the absolute and relative stability (lines 238 – 257).

We agree that crop yields are determined by soil, climate and resource inputs, and that increased resource inputs will thus lead to increased yield, which is also clear in the effect on yield through increased N (e.g. Figure 4). However, the effects of inputs like N and P on stability are not that clear, as opposed to positive effects on stability through crop protection inputs. As in most cases we did not detect significant effects on absolute stability, we believe that the increasing effect on relative stability is due to a change in yield level.

Unfortunately, the original datasets, which were used for the analysis, do not contain more detailed variables, that would allow for a further disentangling of the causing effects on stability. We believe that we best used the available data and that it is necessary for future research to further investigate the causes for changes in stability. We also stressed this issue again in the concluding section.

2. Yield stability for single crops at the field level is not necessarily relevant for farm economy or food security.

The authors state a number of times (lines 19, 28, 31 in the Abstract, lines 69, 71) that yield stability (for crops on the paired plots/fields in the data set) translates into food supply predictability, reliable food production, or even global food security. However, this scaling up is in my view not allowed without much more complicated assessments. First, the four crops may be bulk crops, but not the necessarily the ones that confer food security for many people and farmers globally. The food system does not work like that. So, what is to be stabilised – individual crop yields in a variable world, or food provision? And at which scale?

RESPONSE: Thanks for mentioning this important issue. In our new analysis we now included all crops that were available in the original datasets to broaden relevance of our analysis. We have now also largely removed conclusions and statements related to global food security and reliable food production. We also discuss the issue of scale (see line 398-412).

On the farm level, one of the main advantages of using crop rotations is to counter single-crop variability effects on farm income or food production. Crops vary, often asynchronously, in their responses to the environment – weather, pests, etc. – and to economic price volatility, and at the farm level it is the aggregate result on farm economy of the DIFFERENT crops grown that impacts farm performance and food production. If wheat fails, or maize, then beans or vegetables may make up for it. Field scale measures like the ones in this MS are only appropriate at the field scale, but not for farm economy unless farmers are indeed using only one or two crops. But real OA and CA farms usually grow more crops, and actually crop diversity is often higher on organic farms. Therefore the field-scale measures are not proper for comparisons at the farm level. At the farm level, diversification is likely to confer stability, but this is not addressed at all.

RESPONSE: We agree that this was not clear. We addressed the issue of farm vs field scale in the new version (lines 417-420) and also note that our analysis is limited to the field scale and is thus limited in its generalization.

On a regional, national or global level, the simple scaling up is even less appropriate, for both agronomic/ecological and economic reasons. This scaling up requires a completely different approach, that utilises field, farm and regional data in novel ways. For example, fewer crops (in CA) and farm specialisation may entail increased regional synchrony, increasing the risk of regional crop failures because of regional climate or pest/disease outbreaks (examples are e.g. wheat in Australia 2006, 2009, Russia 2010, etc). This is something that needs to be included in such scaling-up assessments. Diversification would be a means to counter such effects at all scales, but the present analysis is not useful for these questions, not even as a background.

RESPONSE: Thanks for mentioning this issue. We have included a discussion on the necessity and advantages of diversification in the manuscript (lines 398-416). We also mentioned that scaling up may require a different approach citing the useful comments to reviewer provided (see line 412-416).

Hence the authors should delete all simplified references to food security, only acknowledge that analyses of farm, national and global food security and farm economy need field-scale measures to understand how farmers and policies can enhance stability of food (not the four crops) provision. They should clearly state that this analysis is for plot/field scale paired comparisons. And nothing else.

RESPONSE: We thank the reviewer for this constructive and useful comments. We have now deleted all simplified references to food security and discussed that further analysis at the farm scale and at regional and global level should complement such data (see line 398-416). We also point out that food security and reliable food production is influenced by other factors as well (see line 417-420). Although we agree that farm level measurements are necessary and important, we would like to point out that our analysis, and especially the observed reduced yield stability of OA, is robust among most of crops analysed. Hence, these results most likely will also translate to the farm level because lower yield stability of many crops under OA cannot be solved by having a more diverse crop rotation. Also, our analyses focused on bulk crops that are widely planted in the world (the four crops for which we obtained enough data – e.g. barley, maize, wheat and soybean - are among the ten most widely produced staple food crops in the world according to the FAO).

3. Yield stability in highly productive systems is bought with increased external inputs and at the expense of the environment.

Assuming that CV does measure yield stability, it would be relevant to ask what it is in CA that makes yields more stable than OA. The authors touch on this several times but not enough, in my view. Conventional farming uses more inputs than OA, from energy (N-fertilisers, machinery) and fertilisers (e.g. P, K, micronutrients), to pesticides, herbicides, fungicides, etc. (but OA is not innocent as regards machinery and various types of organic manure). So the "input footprint" is likely larger for CA. Several recent meta-analyses and reviews (e.g. Seufert et al. 2017, Reganold & Wachter 2016, Tuck et al. 2014) show that OA - on average – performs better when it comes to environmental impacts and biodiversity, although exceptions can be found and N-leaching remains a problem for all farming. This means that the "output footprint" may also be larger for CA. Inputs in CA thus have the effect of insuring yields, i.e. increasing yield stability, at the expense of resource use elsewhere. Were there any estimates of nutrient leaching or other input-outputs in the studies used?

RESPONSE: We agree with the reviewer that yield stability in highly productive systems is bought with increased external inputs and at the expense of the environment. We now also stress this in the abstract and further discussion is added on issues related to the environmental effects (e.g. input and output footprint – see line 297-299). We address this clearly (e.g. line 281-300). Note however that the paper focuses on yield stability and due to space limitations we could not extensively discuss this. Note that the data-sets we used to calculate yield stability did not contain sufficient information to calculate the input footprint of the different experiments used for the meta-analysis and as such the literature we cite in the discussion is based on other studies

The consequence of this for sustainability is still debated, but the authors hardly touch on these issues. Is the increased field level single crop stability in developed countries really sustainable, or is its lack of sustainability bought at the expense of excluding today's resources from developing countries today and tomorrow's food production globally? Is it

causing large environmental effects, like nutrient loss, or biodiversity loss? Is it causing effects far away in distant countries, like water eutrophication, climate change or land use changes? CA seems to use less local resources, which means a larger footprint globally. Which systems are likely to be the most resilient to environmental, economic or social perturbations? A balanced discussion on the environmental effects of managing intensified agriculture for yield stability is needed – not long, but addressing the larger-scale implications of the increased yield stability that managing for yield quantity seems to lead to.

RESPONSE: We now further expanded our discussion on issues related to the environmental effects of managing intensified agriculture for yield stability (see above).

Comments on the text:

Lines 73-81. A more in-depth discussion on CV is needed (see above). In the present study, SD (S) does not vary with mean yield (X) so the assumption that CV compares different levels of variation is not really fulfilled. For biological data, it is not clear if the assumption should hold, in fact S (and hence CV) can vary in different ways with X, and this should be examined in more detail before the measure is used (note also the corrected CV formula above).

RESPONSE: A more extensive discussion on CV is added (see above; see Supplementary notes)

Lines 89-92. The measure of yields on fields does not account for the fact that OA can counter field and crop yield variability on the farm level. This study is only relevant for field level assessments, but at farms and landscapes it is only relevant if climate, weeds and pest outbreaks are synchronous over larger scales AND this is not countered by crop diversification and crop rotations. So the authors should think a bit more about the possibility to scale up.

RESPONSE: We now included many more crops in our analysis (see above). Our analysis holds for many crops and hence it is unlikely that having a more diverse crop rotation (which is often the case for OA compared to CA) will increase yield stability of OA.

Line 96. Sentence should be "yield stability in fields of single crops of CA was ... higher".

RESPONSE: the sentence was changed (see line 217-218).

Line 105-107. This statement is unfounded. The confidence intervals for yield stability for the 4 crops highly overlap, so there is NO SIGNIFICANT difference between maize and the other crops in Fig. 1. The only thing that can be said is that maize was not different from 1, but the other crops showed lower stability in OA. The authors need to check this all over the MS, differences between crops and treatments are examined by looking for overlaps among the crop/treatments confidence intervals, not by looking at whether the CI:s differ from 1 or not. This is basic statistical knowledge, so please check carefully.

RESPONSE: we have now altered the text and do not directly compare the crops with each other (as we did before).

Line 126-27. The test of fertilisation regime is good! But the increase in stability with intensification does indeed suggest that stability is bought with external inputs, i.e. there are environmental and societal COSTS of increasing yield stability.

RESPONSE: We state (line 245-246): This indicates that the increased relative stability of conventionally agriculture is, in part, due to higher fertilisation levels and related to the higher yield

Line 134-144. It is good that it is noted that differences in CV are mainly due to differences in mean yield, but the authors should consider its implications. Also, is this effect visible also in CA only or OA only – i.e. plots of CV against X (or logCV vs logX) should be negative also within farming systems. Is this the case? If so, this suggests something interesting and potentially general about increasing yields and yield stability. If not, it would be interesting to understand why.

RESPONSE: we have now tested whether variability is related to yield (see supplementary figure 8). We also extensively discussed about this. This was indeed highly relevant for the analysis.

Line 145. Sentence should read "cause for increasing field scale yield stability" ...

Line 153-57. The point on the lack of breeding is important and should be kept.

Lines 159-196. I wonder if these results, which are interesting, can be scaled up to farm and regional scales. In any case, the analysis suffers from the same problems as discussed above.

RESPONSE: this is now discussed (see above; line 398-416).

Line 181-82. See comment concerning lines 105-107. Maize does not differ from the other crops! Correct in next version!

RESPONSE: We have now removed this.

Line 207. Add that breeding for resistance or competitive ability vs. weeds may also be needed, as dicussed above.

RESPONSE: This is discussed (see line274-280)

Line 234- ... Conclusions. May need changes after the problems above have been addressed.

RESPONSE: We have rewritten large parts of the manuscript.

Materials and Methods

Line 255. One could argue that because the systems differ in inputs, the treatments are in fact not comparable. Any kind of comparison of OA-CA or NT-T entails comparing systems that differ in inputs. Needs a short discussion-

RESPONSE: We address the effects of differences in nutrient inputs and how that affects the comparison betweenOA and CV (see line 238-259).

Line 263-64. This is sensible, but I lack a discussion of how the small sample sizes affect the precision of the CV estimates, how this can be estimated and how it should be included in the analysis. Note also that environmental variation increases with duration, which should (could) be accounted for with e.g. analysis of residuals from a yield stability against time

regression, rather than just the CV or S. However, since the data seems to be paired, this is only important if these relations really differ between CA and OA, or NT and T.

RESPONSE: We did not observe that the variation increases with the duration of the study (see above, Figure 1). Hence, because we did not observe a relationship between the variance and duration of the study, we do not see the necessity to correct for such a relationship. Furthermore, our observations are always paired and the observations per comparison are always based on the same number of years.

Line 294-95. Does this mean that data were indeed missing and that it was not a measured 0 (zero) because of crop failure?

RESPONSE: the data were indeed missing. Hence, we could not include them

Line 308-310. Give the exact formula used, please.

RESPONSE: All these formulas are included (see Material and Methods and also supplementary methods).

Line 311-317. This is a most confusing paragraph. In the Pittelkow et al. paper, I think they were only examining mean yields, so this procedure weighed longer studies higher because of their higher precision. This COULD make sense for yield stability too, but since there are other issues like the increased environmental variation with time, the rationale behind this weighing becomes obscure to me. Why not examine the relations of yield stability vs. duration first, so see if there is any relation that is negative (indicating environmental variation effects) and then re-consider what the weighting should be. You need to think about this very carefully, because weighing longer studies more implies that you confound environmental variation with precision, and thus give more weight to studies with larger environmental variability. Is this what you want?

RESPONSE: Thanks for this. We performed a completely new analysis and the weighting is done based on sampling variances following Nakagawa et al.).

Line 322-324. Does this really mean that plot-n=1 for most of the OA-CA comparisons?? This just can't be the case?! It would mean that MOST comparisons are UNREPLICATED, which is such a dismal finding that it should be highlighted. Or did you just not bother to look for the within-study replication?

RESPONSE: A weighting by the number of replicates was not possible for the dataset on organic farming because the values in the Ponisio et al. dataset were not properly extracted and hence we could not use them. The number of replicates is not anymore included in the new analysis.

Figure 1. Barley differs in yield ratio from Maize and Soybean, but there is no crop difference in stability ratio. You should also add the SD comparison here.

RESPONSE: For all crops we present the relative (e.g. C.V.) and absolute stability (e.g. SD)

Fig. 2. More N and similar N differ in yield ratio, but no differences with N input in stability ratio.

RESPONSE: We observed a significant difference in relative stability if conventional fields received more N (see new figure 4)

Supplementary materials:

Fig S2. (c) OA is NOT more variable in 1/SD since the CI:s overlap 1 in all cases.

RESPONSE: *We now clearly address in the new text whether there significant differences*

Fig S3 (c) Again, OA is NOT more variable in 1/SD.

Fig S4. It's interesting to see this figure, because it highlights how you may use different measures of stability. As argued in McArdle et al, SD is actually also useful to measure stability – it depends on what you are looking for and the structure of the data. You assume that relative variation is the important thing = CV. But why? In this particular study, you can use CV, S or range from the shown data. You could also look at risk of crop failure (but that's not really happening; risk for economic loss could perhaps also be a relevant measure). In any case, CV varies, but S does not, and range not really (CA \approx 4.5, OA \approx 4.8, hardly significant). What does this mean? That measures are not showing consistent results, and THE CHOICE that YOU make on what is interesting is important. I don't think you make a good argument for your choice of CV on lines 76-80 or 134-140, you just uncritically use it. If you still want to use CV (which is OK) the pros and cons should be clearly discussed (see above).

RESPONSE: *We now present the relative and absolute stability and discuss what these measures show. We now discuss that it is important to measure both types of variability (see 354-378), see also above.*

Fig S7. Explain the +RR and +CR in the legend. The reader will have forgotten them.

RESPONSE: *This figure has been removed (see Figure 6, main document)*

Table S2. What does the column "Comparison" mean? Explain.

ALSO: This table should (when/if published) contain all the data used (it is electronic) and include n-values for the treatments too.

RESPONSE: *This is now explained. "Study" and "comparison" refer to the original data-set by Ponisio et al (see new Supplementary Table 3)*

General final words:

I liked this paper a lot, despite my critical comments. It highlights the complexities involved in discussions on food and yield stability, security and – of course – food distribution locally, regionally and globally. There are many problems that need the inputs from this (and other) analyses, in order to get these discussion on the right track. Issues that are important are, e.g.:

- Spatial and temporal scales affecting stability and variability.
- Crop distributions and crop rotations; what should be stabilised, individual crops or food production in a variable world?
- The confusion between food security and local crop stability.
- How do we measure yield stability? What are the drivers of variation in yields and yield stability – agronomically, ecologically, economically? Does economy matter for subsistence farmers? Should we minimize the risk of individual crop and total food production failures?
- Can we compare systems that differ in the amount of inputs and their environmental effects, and if so how?

These are interesting areas for future research, so I hope you can address my concerns above so that discussion on these important issues can progress.

RESPONSE: Thanks for all these suggestions. We have discussed several of the issues mentioned above including food security and local crop stability, the influence of crop rotation, the effects of fertilization level and environmental issues. We did not discuss the economic implications because this falls out of the scope of our study and we also do not have sufficiently space to discuss this.

Jan.bengtsson@slu.se

Reviewer #2 (Remarks to the Author):

A meta-analysis of yield stability in conventional, organic and conservation agriculture

General comments

The topic of yield stability is been discussed frequently among scientists and therefore the meta-analysis of yield stability in various cropping systems is timely. The authors used a subset of two already published data sets, which were reported/published earlier to compare yield between organic and conventional farming systems and to determine yield differences between no-till (including conservation ag) and tilled systems. These two publically available data sets were used here to calculate yield stability.

My main concern with the analysis is how yield stability was calculated. The key issue is that stability is calculated as the ratio of the mean to variability around that mean. This has the effect of skewing the highest yielding system toward higher stability. One way to look at this is by re-ordering the equation, which reveals that to have equal stability of the two systems, a lower yielding system has to have lower variability around the mean. By this definition, a high yielding but highly variable system can have the same "stability" as a lower but consistent yielding system. The clearest example is Fig. S4. Comparing the two systems, they appear to have roughly the same variability around the mean (conventional might even be slightly more variable). However, the conventional system has higher yield stability merely because it has a higher yield.

RESPONSE: Many thanks for your comments and addressing the issue related to the measure of yield stability and how it is affected by yield level. We feel that we did not address this issue sufficiently in the first version. To address this issue, we now introduced the terms absolute stability (measured by standard deviation) and relative stability (measured by coefficient of variation), which therefore measures stability per unit yield produced. The later indeed is affected both by the variability and the mean yield level. We therefore now also included the absolute stability, and treated both measures as equally important. We investigated the relation of both measures to the mean yield level, and included a supplementary note on this issue. Furthermore, we stress that it is important to distinguish between these two concepts (e.g. something which often has not (or only shortly) been done in other papers studying stability (for instance in relation to ecosystem functioning).

As we overall found no or little difference in absolute stability (SD), we also show and discuss that the difference in relative stability is mainly due to the difference in mean yield level, as described by the reviewer.

However, we'd like to stress that besides the argument of the reviewer against the usage of CV and our finding that the difference in relative stability is mainly due to difference in mean yield, relative stability has a meaningful interpretation, as the level of variability is put in relation to the mean yield level.

Should the real issue here be yield stability or should it be yield resilience? Yield stability is about consistency, but are we concerned if there is a year with an above average yield? A year with higher yield equals to higher variability in yield, and therefore lower stability. Should we try to avoid years with poor yield, which involves resilience to disruptions? High variability will be OK when the outcome is higher yield.

Therefore, should the analysis/manuscript focus on resilience in yield and not on stability in yield as defined here?

RESPONSE: We have focused on yield stability as we feel this is, beside mean yield, an important measure to investigate. In this respect, we follow a range of recent papers on this issue, both related to agricultural systems and natural systems (e.g. Deguines et al. 2014; Hautier et al. 2015; Lesur-Dumoulin et al. 2017; Tilman et al. 2006;). We do however agree that resilience is very important for future investigations and as such we discussed this (see line 417-420). However, it was not the goal of this paper to focus on resilience and this would result in a completely different paper.

A second but less of a concern is the conclusions drawn from the tillage data set. Did Pittelkow et al come to the same conclusion as stated here on lines 173-175? Likewise, did Pittelkow et al. already concluded what is stated on lines 178-181? In general, when using a data set already analyzed/published, the authors who are (re)-using this data set should be careful to report what is a new finding or what is merely stating again what has already been reported in the earlier publications.

RESPONSE: In our new analysis, the observed yield difference, varied slightly from the results in the original analysis (2% yield difference vs 5.7%). We included a discussion on the possible reasons for this reason (reduced dataset and different modelling approach accounting for the hierarchical structure in the dataset). Moreover, Pittelkow et al. focused on yield differences while our manuscript focuses on yield variability. We have largely rewritten the manuscript and feel that we made clear which conclusion were based on Pittelkow et al. and our analysis.

Specific comments:

Line 101. Wheat does not show a significant difference in Fig. 1 as it crosses the 0 line.

RESPONSE: This is changed now

Line 139: Why is there a reference here to Fig. 1? Are the supporting data only shown in Fig. S2?

RESPONSE: The new figure contains all the data

Reviewer #3 (Remarks to the Author):

Heijden and Knapp compare the yield stability of organic, conventional and no-till agriculture of major cereals crops. I have three major concerns:

1) Without reading the methods in detail, the article appears to compare cropping systems broadly, when only four crop species had enough data to be compared. Throughout the article (in the abstract, title, etc.) that this is a comparison of only four crops (though major important crops) should be made clear.

RESPONSE: Many thanks for your comments. We have now reanalyzed the data-set based on your comments and those of the other reviewers. In order to widen the scope of the analysis, we now included all crops that were available in the original datasets and made it clear in the text, where only less crops are compared.

2) The coefficient of variation, when applied to a small or moderately sized sample, tends to be too low. The corrected version should be used:

$$cv^{\wedge} = (1 + 1/4n) * cv$$

RESPONSE: Thanks for the suggestion of this correction. Reviewer 1 made the same comment. We were not aware of this. While this correction factor is of particular importance

when comparing variation (measured by CV) between treatments or species, that are based on a different (and small) number of observations per treatment or species, in our analysis we always compare the same number of observations (i.e. years) for both treatments. Furthermore, as our analysis is based on pairwise ratios of the CVs of both treatments and each pairwise observation has the same number of years of observation (n) this correction factor cancels out due to:

$$CVR = \frac{CV_{exp} * (1 + \frac{1}{4n_{exp}})}{CV_{cont} * (1 + \frac{1}{4n_{cont}})} \text{ with } n = n_{exp} = n_{cont} \text{ simplifies to } CVR = \frac{CV_{exp} * (1 + \frac{1}{4n})}{CV_{cont} * (1 + \frac{1}{4n})}$$

and thus $CVR = \frac{CV_{exp}}{CV_{cont}}$.

We did therefore not use this correction factor in our new analysis.

3) The authors follow the analytic methods of Pittelkow et al. These methods do not account for the fact that multiple CV are extracted from the same study. They also do not account from random variation between studies. Like in Seufert et al. this study thus suffers from pseudo-replication which inflates the type I error. The authors should re-analyze this data set using a random effects meta-analytic model:

$$y_{ij} = \mu + \alpha_i + e_{ij}$$

$$\alpha_i \sim N(0, \sigma^2_{\alpha})$$

$$e_{ij} \sim N(0, S_{ij})$$

where y_{ij} is the observed magnitude of the j th cv^* of the i th study, μ is the mean cv^* across the studies, α_i is the effect of i study, and e_{ij} is the residual. σ^2_{α} is the between study variance, and S_{ij} is the variance of cv^*_{ij} . The funny thing here is that the cv is a cv of means, so it is not immediately clear to me what estimate should be used. Perhaps the SE of the CV would be sufficient? $cv/4n$. It would be best to come up with a clever way to incorporate the reported SD around the mean yield for each year reported in the studies.

RESPONSE: Thanks for the suggestions. We conducted a completely new analysis employing the suggested mixed model including a random study effect. As responses we used the mean yield ratio, absolute stability ratio (which is the ratio of the SDs over years), and relative stability ratio (which is the ratio of the CVs across years). Regarding the sampling variances (S_{ij} in your notation) for the different responses we followed the approach proposed by Nakagawa et al (2015), where equations for the sampling variances for absolute stability (SD) and relative stability (CV) were derived. Here, the sampling variance for the relative stability is dependent on the number of years, the mean yield and the SD over years. We used these sampling variances as they were implemented in the R package metafor (see also the respective section in the Supplementary Methods). As the error measure (e.g. SD) of the mean yield of each year (as suggested by the reviewer) was not available in the dataset on no-tillage and only sparsely in the dataset on organic farming, we could not employ the suggested procedure.

Also, what was done when multiple organic treatments were paired with the same conventional control? (i.e., a multi treatment experiment in the original study). This is not clear from the methods.

RESPONSE: In the first analysis, we did not correct for this fact. However, in our new analysis using a mixed-model approach, we included a variance-covariance structure to correct for this issue. For the response mean yield ratio, we used the suggested correction from Lajeunesse (2011) which was also employed in the original analysis of the dataset on organic farming (Ponisio et al. 2015). For the two stability responses we derived the

respective matrices and also validated them through a small simulation. We have added a section, also containing the matrices in the Supplementary Methods.

After the authors re-analyze the data I would like to read the manuscript again because many results may change.

Reviewers' comments:

Reviewer #1 (Remarks to the Author):

This is a highly improved version of a previously submitted manuscript. The authors have added several analyses and results that improve the basis of the interpretations they make, and have re-interpreted some of the results in a good way. The MS reads well, although the Results and Discussion section is a bit long and can be shortened. However, all the issues in the section need to be discussed, so it's difficult to shorten it as reviewer. But the authors may consider which parts can be shortened by a sentence or two, or rewritten in a more succinct way.

However, there are a number of minor issues that must be addressed:

Line 41. (a) "Productivity" is (in my view, and among ecologists in general) an inherent property of the site and its environmental conditions, based on naturally given resource levels and climate – and perhaps including inputs like N, P or other nutrients. But what the authors mean here (and also elsewhere in the MS) is actual production measured as yield. So the word production (or yield) should be used. Or am I wrong?

(b) Production has NOT tripled the last decade (10 years), but rather during the 20th century, last century or since the 1950-ies. This is what the referenced authors 1-3 more or less refers to, and the authors need to check that they get this sentence right.

Line 212 and 213. Both sentences start with "This". Avoid this.

Line 231. In Figure 1, the two crops that have higher yield stability in CA are soybean and barley (not maize as written in the text).

Line 260-261. The addition of green manure in OA does NOT SIGNIFICANTLY enhance yield or relative stability (compared to CA). This would have been indicated by 95% confidence intervals not overlapping between the "with green manure" and the "without green manure" – indicating a significant difference between them. But that is not so in the Suppl. Figure 5; the CI:s overlap in all cases, meaning there is not significant difference. So this passage has to be reworded.

Lines 286-292. The authors make the mistake (in my opinion) to believe that a global strategy is needed (interpreted by me as a single global strategy to improve yields and stability). They are confusing "global" with western, intensive agriculture in developed countries as being applicable globally. But the low-intensity, low soil fertility agriculture in developing countries may need a different strategy than the Euro-American agriculture where most of the organic farming studies have been made. From a sustainability and climate mitigation perspective, improving yields in Europe – according to the intensification paradigm – may not be a clever idea, since any fixed amount of inputs (be it fertilisers or technology) will have a much higher pay-off or effect on yield in developing countries where people would profit immensely from a 50-100 % increase in yields with simple measures, while the same kg of fertilisers (for example) will have a much lower effect in the already richer countries.

I am sure the authors are well-meaning, but this passage is too much embedded in the intensification narrative to be credible as a "global" strategy. Sorry, but that's my strong opinion (but it's based on biophysical facts).

Line 304. Since the sentence already uses the word "reduces", the minus (-) signs are not needed before 2, 1 and 4%. (similarly on line 332).

Line 331 and 333. Both sentences start with "However".

Line 356. The text should be (cf. above) "corrects for the level of yield ..." not productivity.

Line 363. I think it is a bit strange to call the dataset "your" data as it was collected by other authors. Why not say "in the data we have analysed" or something more neutral.

Line 378-79 or thereabouts. Here I asked myself, what is it that a farmer or farm economy would worry about? Is it CV or absolute stability? Is it on field or farm level? Does it depend in the cropping systems used, and on how different crops respond to weather and environmental variation? Or price volatility? It seems to me that the discussion is very technical, but it would be nice to think about what the farmers would like to know. Somewhere you need a sentence or two in this issue (see also below on comment to line 405-07).

Line 405-407. I am not convinced by this reasoning. Even if several single crops show reduced stability under OF (or low intensity in general), the crucial thing for a farmer is if total farm output/yield/income is more stable across years. And to discuss this, you need the variability of different crops and an estimate of synchrony in variation. In the ecological diversity literature, the key issue is if different species (crops) have a response diversity to environmental variation or not. If some crops do better under wet, other under dry conditions, for example, then diversity may lead to higher stability at the farm but not monocultural field level. So the key issue is if crops respond synchronously or not to external variation (or even price volatility). Hence the logic on these lines is questionable, and they should perhaps be deleted, if they authors don't want to pursue the line of reasoning I just gave.

Line 420. Why the words "local food security" and not farm income or something like that? Food security in the westernised agriculture is not at all based on "local". Or what do you mean by "local"?

Line 437. Insert a space between "stability" and "of".

Figure 1, legend. What does the p-value refer to? Not the overall model, but probably a full (or reduced?) model with crop type as factor. Be clear about this (also in Figs 4, 5 and 6)

Supplementary materials, lines 79-84.

When discussing if CV is independent on the mean (lines 50-84) I think there is a point of discussion. I believe it is generally agreed that p-levels like 0.05 are only a convention that's been decided on for pragmatic reasons. But really probabilities are a continuum and it's largely a matter of opinion which p-value to choose and if one- or two-tailed probabilities should be used. Which is why I nowadays tend to let the readers and users of the data decide on "significant" when there are borderlines. Here the authors argue that since some regressions of log CV on log Mean are "not significantly different from zero" with $p=0.06$, there is no problem interpreting the CV as independent of the mean. But really $p=0.06$ still suggests something is going on, and since I would have expected (based on Bayesian-style previous knowledge) a negative slope, one could just as well argue that there is such an effect in the data with one-tailed $p=0.03$ (keeping the frequentist approach).

So I am not convinced by these arguments. I am, however, not sure how this would affect the results. But I expect the authors to add a caveat in the Supplementary materials on this issue, and what it would have meant if the CV indeed is related to the mean with a negative slope.

And that's it! A good revision of the previous MS, with just some minor points to amend now.

In my view,

Jan.bengtsson@slu.se

Reviewer #4 (Remarks to the Author):

I was asked to comment on the authors' use of meta-analytic model in place of R3, so I confine the scope of my review to this topic.

The authors appear to have analysed the data using the metafor package, which is what I would have done to fit this model. It is hard for me to comment on whether the model has been implemented 100% correctly without seeing their code, but the written method suggests to me that it is right. I would encourage the authors to state explicitly which function was used – I suspect it was `rma.mv`.

If so, I would also encourage the authors to check the residual error term was explicitly expressed. The function does not do this automatically, which sometimes trips people up; usually this is expressed as random-factor with a level corresponding to each effect size, which then correlates with the VCV matrix that they describe.

Point by Point Response list to the reviewers comments:

Reviewer #1 (Remarks to the Author):

This is a highly improved version of a previously submitted manuscript. The authors have added several analyses and results that improve the basis of the interpretations they make, and have re-interpreted some of the results in a good way. The MS reads well, although the Results and Discussion section is a bit long and can be shortened. However, all the issues in the section need to be discussed, so it's difficult to shorten it as reviewer. But the authors may consider which parts can be shortened by a sentence or two, or rewritten in a more succinct way.

Thank you for your time and valuable suggestions for further improvement. We have shortened the Results and Discussion section at several places (see track changes, e.g. line 272-277; 437-440; 468-473) as recommended. Please note that in the earlier manuscript there was one section with results and discussion. This was not in line with the editorial policy of Nature Communications, and we have now corrected this and a separate results and discussion section are included. As a consequence, several parts have been moved and these are indicated through track changes (these parts in itself are hardly changed with few minor edits).

However, there are a number of minor issues that must be addressed:

Line 41. (a) "Productivity" is (in my view, and among ecologists in general) an inherent property of the site and its environmental conditions, based on naturally given resource levels and climate – and perhaps including inputs like N, P or other nutrients. But what the authors mean here (and also elsewhere in the MS) is actual production measured as yield. So the word production (or yield) should be used. Or am I wrong?

Thank you for pointing this out. We have now changed this throughout the manuscript and use the word "yield" or production (or food production)

(b) Production has NOT tripled the last decade (10 years), but rather during the 20th century, last century or since the 1950-ies. This is what the referenced authors 1-3 more or less refers to, and the authors need to check that they get this sentence right.

This has now been changed. Thanks for pointing this out.

Line 212 and 213. Both sentences start with “This”. Avoid this.

Changed

Line 231. In Figure 1, the two crops that have higher yield stability in CA are soybean and barley (not maize as written in the text).

Changed. Thanks for pointing this out.

Line 260-261. The addition of green manure in OA does NOT SIGNIFICANTLY enhance yield or relative stability (compared to CA). This would have been indicated by 95% confidence intervals not overlapping between the “with green manure” and the “without green manure” – indicating a significant difference between them. But that is not so in the Suppl. Figure 5; the CI:s overlap in all cases, meaning there is not significant difference. So this passage has to be reworded.

The sentence has been rephrased (see line 271-277). We meant to say that the addition of green manure in organic agriculture has a positive impact on the relative yield stability (e.g. green manure reduced the relative yield instability in organic agriculture)

Lines 286-292. The authors make the mistake (in my opinion) to believe that a global strategy is needed (interpreted by me as a single global strategy to improve yields and stability). They are confusing “global” with western, intensive agriculture in developed countries as being applicable globally. But the low-intensity, low soil fertility agriculture in developing countries may need a different strategy than the Euro-American agriculture where most of the organic farming studies have been made. From a sustainability and climate mitigation perspective, improving yields in Europe – according to the intensification paradigm – may not be a clever idea, since any fixed amount of inputs (be it fertilisers or technology) will have a much higher pay-off or effect on yield in developing countries where people would profit immensely from a 50-100 % increase in yields with simple measures, while the same kg of fertilisers (for example) will have a much lower effect in the already richer countries.

I am sure the authors are well-meaning, but this passage is too much embedded in the intensification narrative to be credible as a “global” strategy. Sorry, but that’s my strong opinion (but it’s based on biophysical facts).

We have now replaced “global” with “multi-faceted” (see line 393) to make clear that there are a range of different strategies and we agree that strategies for western intensive agriculture are different compared to that in some developing countries.

Line 304. Since the sentence already uses the word “reduces”, the minus (-) signs are not needed before 2, 1 and 4%. (similarly on line 332).

Changed

Line 331 and 333. Both sentences start with “However”.

Altered

Line 356. The text should be (cf. above) “corrects for the level of yield ...” not productivity.

Changed into “production”

Line 363. I think it is a bit strange to call the dataset “your” data as it was collected by other authors. Why not say “in the data we have analysed” or something more neutral.

Changed

Line 378-79 or thereabouts. Here I asked myself, what is it that a farmer or farm economy would worry about? Is it CV or absolute stability? Is it on field or farm level? Does it depend in the cropping systems used, and on how different crops respond to weather and environmental variation? Or price volatility? It seems to me that the discussion is very technical, but it would be nice to think about what the farmers would like to know. Somewhere you need a sentence or two in this issue (see also below on comment to line 405-07).

Line 405-407. I am not convinced by this reasoning. Even if several single crops show reduced stability under OF (or low intensity in general), the crucial thing for a farmer is if total farm output/yield/income is more stable across years. And to discuss this, you need the variability of different crops and an estimate of synchrony in variation. In the ecological diversity literature, the key issue is if different species (crops) have a response diversity to environmental variation or not. If some crops do better under wet, other under dry conditions, for example, then diversity may lead to higher stability at the farm but not monocultural field level. So the key issue is if crops respond synchronously or not to external variation (or even price volatility). Hence the logic on these lines is questionable, and they should perhaps be deleted, if they authors don't want to pursue the line of reasoning I just gave.

This is an important and interesting discussion point. We have now removed the statement that the enhanced crop diversity in organic farming may not be sufficient to reduce yield instability in organic farming (see line 461-466). With the available data-set it is indeed not possible to estimate the synchrony of yield stability (or instability) in different crops and it is difficult to make any predictions.

Line 420. Why the words “local food security” and not farm income or something like that? Food security in the westernised agriculture is not at all based on “local”. Or what do you mean by “local”?

We have now changed this into: “the ability of a particular system to produce enough food or income.” (see line 485)

Line 437. Insert a space between “stability” and “of”.

Changed

Figure 1, legend. What does the p-value refer to? Not the overall model, but probably a full (or reduced?) model with crop type as factor. Be clear about this (also in Figs 4, 5 and 6)

The p-value was from an ANOVA, testing whether there was a difference among crop species (or the factor presented, for the other figures). As we did not refer to those test in the manuscript we decided to remove the p-values from all figures and also removed the respective description from the method section.

Supplementary materials, lines 79-84.

When discussing if CV is independent on the mean (lines 50-84) I think there is a point of discussion. I believe it is generally agreed that p-levels like 0.05 are only a convention that's been decided on for pragmatic reasons. But really probabilities are a continuum and it's largely a matter of opinion which p-value to choose and if one- or two-tailed probabilities

should be used. Which is why I nowadays tend to let the readers and users of the data decide on “significant” when there are borderlines. Here the authors argue that since some regressions of log CV on log Mean are “not significantly different from zero” with $p=0.06$, there is no problem interpreting the CV as independent of the mean. But really $p=0.06$ still suggests something is going on, and since I would have expected (based on Bayesian-style previous knowledge) a negative slope, one could just as well argue that there is such an effect in the data with one-tailed $p=0.03$ (keeping the frequentist approach).

So I am not convinced by these arguments. I am, however, not sure how this would affect the results. But I expect the authors to add a caveat in the Supplementary materials on this issue, and what it would have meant if the CV indeed is related to the mean with a negative slope.

We fully agree that the common 0.05 threshold is only a convention. We made it therefore clear that this level is used as significance threshold here, and added a discussion on this “slight significance” and the meaning of a negative relationship (see Supplementary notes, line 81-91). We, furthermore, discuss the implications of the absence of the Taylor Power Law here, and the necessity of careful interpretation of CV.

And that's it! A good revision of the previous MS, with just some minor points to amend now.

In my view,
Jan.bengtsson@slu.se

Reviewer #4 (Remarks to the Author):

I was asked to comment on the authors' use of meta-analytic model in place of R3, so I confine the scope of my review to this topic.

Thank you for time and constructive evaluation of the meta-analytic part and your time to review this manuscript.

The authors appear to have analysed the data using the `metaphor` package, which is what I would have done to fit this model. It is hard for me to comment on whether the model has been implemented 100% correctly without seeing their code, but the written method suggests to me that it is right. I would encourage the authors to state explicitly which function was used – I suspect it was `rma.mv`.

We indeed used the `rma.mv()` function from the `metaphor` package and added this information to the manuscript (see line 202).

If so, I would also encourage the authors to check the residual error term was explicitly expressed. The function does not do this automatically, which sometimes trips people up; usually this is expressed as random-factor with a level corresponding to each effect size, which then correlates with the VCV matrix that they describe.

Thanks for pointing this out. We did check whether the residual error term was explicitly expressed. Our analysis in R consisted of three major steps:

- 1) A function to calculate and add the VCV matrix containing the corrected sampling variances, which was used as provided by Lajeunesse 2011 and modified to also calculate the respective matrices for the two stability measures as shown in the Supplemental Material.
- 2) Adding the respective response measure using the `escalc()` function from the `metaphor` package (it already includes the measures VR (for our absolute stability) and CVR (for our relative stability), while the latter differs from the calculation in Nakagawa et al. 2015, as described in the method section).
- 3) The actual model (shown for the overall effect containing only an intercept) using the `rma.mv()` function with the following code:

```
model <- rma.mv(yi~1, V=varmat, random=~1|study/myo, data=metadat)
```

where `yi` is a column in the metadata and contains the respective ratio (effect size) as calculated by `escalc()`; `varmat` contains the VCV matrix with the sampling variances; `study` is a column in the metadata encoded as a factor representing the random study effect, and `myo` is a column in metadata which denotes the single observation and thus represents the residual error term. We therefore included the residual error term explicitly as a random factor, as requested.

The default fitting method is REML, and thus needs not to be specified.

In order to test the effect of moderators (as described in methods), the code for the model without intercept to get the means of each factor level is then:

```
model <- rma.mv(yi~modi -1, V=varmat, random=~1|study/myo, data=metadat)
```

where `modi` is a column in the metadata encoded as factor and denoting the moderator to test.

REVIEWERS' COMMENTS:

Reviewer #4 (Remarks to the Author):

The authors have addressed all of my former concerns, and I believe that their analysis is appropriate.